



# Improved NMR transfer of magnetization from protons to half-integer spin quadrupolar nuclei at moderate and high MAS frequencies

Jennifer S. Gómez[1], Andrew G.M. Rankin[1,#], Julien Trébosc[2], Frédérique Pourpoint [1], Yu Tsutsumi[3], Hiroki Nagashima[4], Olivier Lafon[1,5], Jean-Paul Amoureux[1,6,7]

[1]Univ. Lille, CNRS, Centrale Lille, Univ. Artois, UMR 8181 – UCCS – Unité de Catalyse et Chimie du Solide, Lille, 59000, France
[2]Univ. Lille, CNRS, INRAE, Centrale Lille, Univ. Artois, FR 2638 – IMEC – Fédération Chevreul, Lille, 59000, France
[3]Bruker Japan, 3-9 Moriya-cho, Kanagawa-ku, Yokohama-shi, Kanagawa, 221-0022, Japan
[4]Interdisciplinary Research Center for Catalytic Chemistry, National Institute of Advanced Industrial Science and Technology (AIST), 1-1-1 Higashi, Tsukuba, Ibaraki, 305-8565, Japan
[5]Institut Universitaire de France, 1 rue Descartes, Paris, 75231, France
[6]Riken NMR Science and Development Division, Yokohama-shi, Yokohama-shi, Kanagawa, 230-0045, Japan
[7]Bruker Biospin, 34 rue de l'industrie, Wissembourg, 67166, France
[#]Present address: Sorbonne Université, CNRS, Collège de France, Laboratoire de Chimie de la Matière Condensée de Paris (LCMCP), 4 place Jussieu, Paris, 75005, France

*Correspondance to*: Olivier Lafon (olivier.lafon@univ-lille.fr) and Jean-Paul Amoureux (jean-paul.amoureux@univ-lille.fr)

**Keywords:** Quadrupolar nuclei, proton, *D*-RINEPT, PRESTO, adiabatic pulses, composite pulses.

**Abstract.** Half-integer spin quadrupolar nuclei are the only magnetic isotopes for the majority of the chemical elements. Therefore, the transfer of polarization from protons to these isotopes under magic-angle spinning (MAS) can provide precious insights into the interatomic proximities in hydrogen-containing solids, including organic, hybrid, nanostructured and biological solids. Furthermore, this transfer has recently been combined with dynamic nuclear polarization (DNP) in order to enhance the NMR signal of half-integer quadrupolar isotope. Nevertheless, the cross-polarization transfer lacks of robustness in the case of quadrupolar nuclei and we have recently introduced as an alternative technique a through-space refocused insensitive nuclei enhancement by polarization transfer (*D*-RINEPT) scheme combining hetero-nuclear dipolar recoupling built from adiabatic pulses and continuous wave decoupling. This technique has been demonstrated at 9.4 T with moderate MAS frequencies, $\nu_R \approx$ 10-15 kHz, in order to transfer the DNP-enhanced $^1$H polarization to quadrupolar nuclei. Nevertheless, polarization transfers from protons to quadrupolar nuclei are also required at higher MAS frequencies in order to improve the resolution of $^1$H spectra. We investigate how this transfer can be achieved at $\nu_R \approx$ 20 and 60 kHz. We demonstrate that the *D*-RINEPT sequence using adiabatic pulses still produces efficient and robust transfer but requires large rf-fields, which may not be compatible with the specifications of commonly employed MAS NMR probes. As an alternative, we introduce robust and efficient variants of *D*-RINEPT and PRESTO (phase-shifted recoupling effects a smooth transfer of order) sequences using symmetry-based recoupling schemes built from single and composite π-pulses. Their performances are compared using the



average Hamiltonians and experiments at $B_0$ = 18.8 T on γ-alumina and isopropylamine templated microporous
aluminophosphate AlPO$_4$-14, featuring low and significant $^1$H-$^1$H dipolar interactions, respectively. These experiments
demonstrate that the $^1$H magnetization can be efficiently transferred to $^{27}$Al nuclei using $D$-RINEPT with SR4$_1^2$(270$_0$90$_{180}$)
recoupling, and PRESTO with R22$_2^7$(180$_0$) or R16$_7^6$(270$_0$90$_{180}$) schemes at $\nu_R$ = 20 or 62.5 kHz, respectively. The $D$-RINEPT
and PRESTO recouplings complement each other since the latter is affected by dipolar truncation, whereas the former is not.


## I. Introduction

Quadrupolar nuclei with a nuclear spin quantum number $S$ = 3/2, 5/2, 7/2 or 9/2 are the only NMR-active isotopes for over
60% of the chemical elements of the first six periods of the periodic table, including six of the eight most abundant elements
by mass in the Earth's crust: O, Al, Ca, Na, Mg and K.(Ashbrook and Sneddon, 2014) A wide range of materials, including
organic compounds, biological macromolecules as well as nanostructured or hybrid materials, contain half-integer spin
quadrupolar nuclei and protons. Proximities between these isotopes have notably been probed in solid-state NMR experiments
by transferring the polarization of protons to half-integer quadrupolar nuclei through dipolar couplings under magic-angle
spinning (MAS) conditions.(Rocha et al., 1991; Hwang et al., 2004; Peng et al., 2007; Vogt et al., 2013; Chen et al., 2019)
More recently, this polarization transfer has been combined under MAS with DNP (dynamic nuclear polarization) in order to
enhance the NMR signals of half-integer spin quadrupolar nuclei.(Vitzthum et al., 2012; Perras et al., 2015a; Nagashima et al.,
2020) This approach has notably allowed the detection of insensitive quadrupolar nuclei with low natural abundance, such as
$^{17}$O or $^{43}$Ca, or low gyromagnetic ratio, γ, such as $^{47,49}$Ti, $^{67}$Zn or $^{95}$Mo, near surfaces of materials.(Perras et al., 2015a;
Nagashima et al., 2020; Blanc et al., 2013; Perras et al., 2016, 2017; Hope et al., 2017; Lee et al., 2017; Nagashima et al., n.d.;
Li et al., 2018)
This transfer has originally been achieved using cross-polarization under MAS (CPMAS).(Harris and Nesbitt, 1988)
Nevertheless, this technique lacks robustness for quadrupolar nuclei since the spin-locking of the central transition (CT)
between energy levels ±1/2 is sensitive to the strength of the quadrupole interaction, the offset, the CSA (chemical shift
anisotropy) and the rf-field inhomogeneity.(Vega, 1992; Amoureux and Pruski, 2002; Tricot et al., 2011, p.20) Furthermore,
CPMAS experiments require a careful adjustment of the rf-field applied to the quadrupolar isotope in order to fulfill the
Hartmann-Hahn condition $(S+1/2)\nu_{1S} + \varepsilon\nu_{1H} = n\nu_R$, where $\nu_{1S}$ and $\nu_{1H}$ denote the amplitudes of the rf-fields applied to the
quadrupolar isotope $S$ and to the protons, respectively, $\varepsilon$ = ±1, $n$ = ±1 or ±2, and $\nu_R$ is the MAS speed, while avoiding the rotary
resonance recoupling (R$^3$) $\nu_{1S} = p\nu_R/(S+1/2)$ with $p$ = 0, 1, 2, 3.(Amoureux and Pruski, 2002; Ashbrook and Wimperis, 2009)
Moreover, the magnetization of the quadrupolar nuclei cannot be spin-locked for some crystallite orientations, which leads to
line-shape distortions.(Barrie, 1993; Hayashi and Hayamizu, 1993; Ding and Mcdowell, 1995)
These issues have been circumvented by the use of the PRESTO (phase-shifted recoupling effects a smooth transfer of order)
scheme,(Perras et al., 2015a, b) and more recently, the through-space refocused INEPT (denoted RINEPT



hereafter).(Nagashima et al., 2020, n.d.; Giovine et al., 2019) These schemes benefit from higher robustness than CPMAS since they do not employ a spin-lock on the quadrupolar channel, but instead a limited number (two or three) of pulses selective to the CT. In these sequences, the dipolar interactions between protons and quadrupolar nucleus are reintroduced by applying

on the $^1$H channel symmetry-based recoupling sequences, such as R18$_2^5$ for PRESTO or SR4$_1^2$ for RINEPT.(Zhao et al., 2001; Brinkmann and Kentgens, 2006a) In the case of recoupling sequences built from single square π-pulses, the RINEPT sequence using SR4$_1^2$ (denoted RINEPT-SR4$_1^2$) is more efficient than PRESTO at $\nu_R \geq 60$ kHz because of its higher robustness to rf-field inhomogeneity and $^1$H offset and CSA. At $\nu_R \leq 20$ kHz, the PRESTO technique is more efficient since the efficiency of RINEPT-SR4$_1^2$ is reduced by the increased losses due to $^1$H-$^1$H interactions at lower MAS frequencies during the SR4$_1^2$

recoupling and the windows used to rotor-synchronize the SR4$_1^2$ blocks, whereas the PRESTO sequence is devoid of these windows.(Giovine et al., 2019a)

Recently, we have introduced a novel variant of the RINEPT sequence employing SR4$_1^2$ recoupling built (i) from tanh/tan (tt) adiabatic inversion pulses, (ii) continuous-wave (CW) irradiations during the windows, and (iii) composite π/2 and π pulses on the $^1$H channel, in order to limit the losses due to $^1$H-$^1$H interactions and improve the transfer efficiency at moderate MAS

frequencies.(Nagashima et al., 2020, n.d.) This novel RINEPT variant, denoted RINEPT-CWc-SR4$_1^2$(tt), is more efficient than PRESTO and CPMAS at $\nu_R \approx 12.5$ kHz and has been combined with DNP to detect the NMR signal of quadrupolar nuclei with small dipolar coupling with protons, including quadrupolar low-γ isotopes, such as $^{47,49}$Ti, $^{67}$Zn or $^{95}$Mo, and unprotonated $^{17}$O nuclei.

However, several NMR experiments require the transfer of $^1$H magnetization to quadrupolar nuclei at $\nu_R > 12.5$ kHz. In

particular, MAS frequencies of $\nu_R \geq 20$ kHz are needed to avoid the overlap between the center-bands and the spinning sidebands of satellite transitions (ST) in $^{27}$Al spectra at 18.8 T. In addition, magnetization transfers at $\nu_R \geq 60$ kHz are advantageous to acquire 2D hetero-nuclear correlation spectra between protons and quadrupolar nuclei endowed with high resolution along the $^1$H dimension since fast MAS averages out the $^1$H-$^1$H dipolar couplings.

Concurrently, we have demonstrated that the efficiency of PRESTO transfers using the R16$_7^6$ recoupling can be improved at

$\nu_R = 62.5$ kHz using $270_0 90_{180}$ composite π-pulses as a basic inversion element, where the standard notation for composite pulses is used: $\xi_\phi$ denotes a rectangular, resonant rf-pulse with flip angle ξ and phase ϕ in degrees.(Giovine et al., 2019a) More recently, SR4$_1^2$ and R12$_3^5$ recoupling schemes built from $90_{-45} 90_{45} 90_{-45}$ composite π-pulses have been proposed, but they have not yet been incorporated into RINEPT transfers.(Perras et al., 2019) Globally, no systematic study of R$N_n^\nu$ recouplings built from composite π-pulses has been carried out.

In the present article, we investigate the use of RINEPT-CWc using an adiabatic recoupling scheme at the higher MAS frequencies of $\nu_R = 20$ and $62.5$ kHz. We demonstrate using numerical simulations of spin dynamics and experiments on γ-alumina and isopropylamine templated microporous aluminophosphate AlPO$_4$-14 (hereafter AlPO$_4$-14) that the rf requirement of this technique increases with the $^1$H-$^1$H dipolar interactions and is not compatible with the specifications of most MAS probes at $\nu_R \geq 20$ kHz. As an alternative, we introduce variants of the PRESTO and RINEPT sequences by selecting with AH





(average Hamiltonian) the recoupling schemes built from single rectangular or composite π-pulses. Finally, using experiments on γ-alumina and $AlPO_4$-14, which feature different $^1H$-$^1H$ dipolar interactions, we identify the most robust and efficient PRESTO and RINEPT transfers at $B_0$ = 18.8 T with $\nu_R$ = 20 and 62.5 kHz.

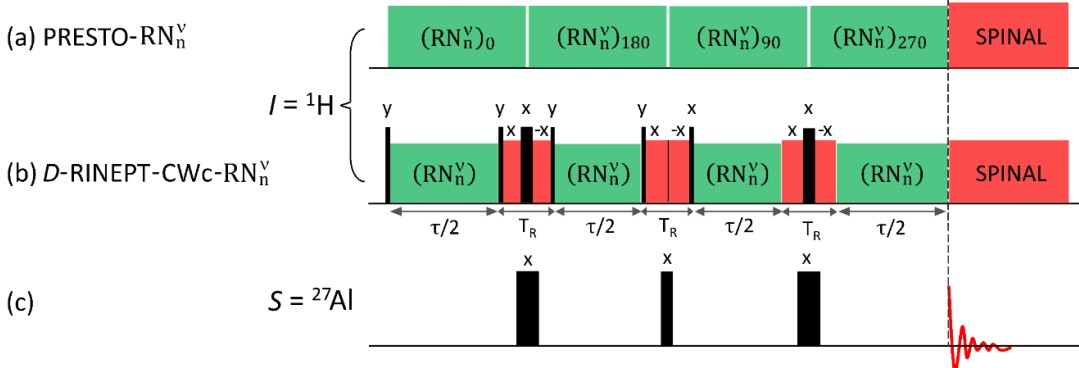

**Figure 1: $^1H \rightarrow ^{27}Al$ (a,c) PRESTO-$RN_n^\nu$ and (b,c) $D$-RINEPT-CWc-$RN_n^\nu$ pulse sequences. The sequences applied to $^1H$ and $^{27}Al$ channels are shown in (a,b) and (c), respectively. The narrow and broad black bars represent π/2 and π-pulses, respectively. The acquisition of the FIDs (indicated with the vertical dashed line) starts after (a) the end of the $RN_n^\nu$ block or (b) on top of the echo shifted with $\tau_R$/2 with respect to the end of the last recoupling block.**

## II. Pulse sequences and theory

**II.1. PRESTO**

**II-1-1. Single-quantum hetero-nuclear dipolar recoupling**

A $RN_n^\nu$ sequence, where $N$ is an even positive integer and $n$ and $\nu$ are integers, consists of $N/2$ pairs of elements $\mathcal{R}_\phi \mathcal{R}'_{-\phi}$ with $\mathcal{R}$ an inversion pulse with a duration of $nT_R/N$, where $T_R = 1/\nu_R = 2\pi/\omega_R$ is the rotor period, $\mathcal{R}'$ an inversion pulse derived from $\mathcal{R}$ by changing the sign of all phases and $\phi = \pi\nu/N$ radians an overall phase shift. The rf-field requirement of $RN_n^\nu$ is equal

to:

$$\nu_1 = \frac{N}{n}\frac{\xi^{tot}}{2\pi}\nu_R \tag{1}$$

where $\xi^{tot} = \sum_{i=1}^{P} \xi^i$ is the sum of the flip angles of the $P$ individual pulses of the $\mathcal{R}$ element.

In the PRESTO sequence (Fig.1a), symmetry-based γ-encoded $RN_n^\nu$ schemes applied to the $^1H$ channel reintroduce the $|m|$ = 2 space components and the single-quantum (SQ) terms of the hetero-nuclear dipolar couplings between the protons and the

quadrupolar nuclei, as well as the $^1H$ CSA, while they suppress the contributions of $^1H$ isotropic chemical shifts, the hetero-nuclear $J$-couplings with protons, and the $^1H$-$^1H$ dipolar couplings to the first-order average Hamiltonian.(Zhao et al., 2004) The hetero-nuclear dipolar interaction is characterized by a space rank $l$ and a spin rank $\lambda$. A γ-encoded $|m|$ = 2 SQ hetero-



nuclear dipolar recoupling must selectively reintroduce the two components $\{l, m, \lambda, \mu\} = \{2, 2, 1, \mu\}$ and $\{2, -2, 1, -\mu\}$ of the hetero-nuclear dipolar coupling and ${}^1$H CSA with $\mu = \pm1$, while other components must be suppressed.

During these recoupling schemes, the contribution of the dipolar coupling between $I = {}^1$H and $S$ nuclei to the first-order Hamiltonian is equal to:(Zhao et al., 2004)

$$\bar{H}_{D,IS}^{(1)} = \omega_{D,IS} S_z [I^+ \exp(i2\varphi) + I^- \exp(-i2\varphi)], \tag{2}$$

where $I^\pm = I_x \pm iI_y$ are the shift operators, and the magnitude and phase of the recoupled $I$-$S$ dipolar coupling are given by

$$\omega_{D,IS} = -\kappa \frac{\sqrt{3}}{2} b_{IS} \sin^2\left(\beta_{PR}^{D,IS}\right) \tag{3}$$

and

$$\varphi = \gamma_{PR}^{D,IS} - \omega_R t^0, \tag{4}$$

respectively, where $b_{IS}$ is the dipolar coupling constant in rad/s, and $\kappa$ is the scaling factor of the recoupled hetero-nuclear dipolar interaction, which depends on the $RN_n^\nu$ symmetry and the $\mathcal{R}$ element. The Euler angles $\left\{0, \beta_{PR}^{D,IS}, \gamma_{PR}^{D,IS}\right\}$ relate the $I$-$S$ vector to the MAS rotor frame, and $t^0$ refers to the starting time of the recoupling. The norm of $\bar{H}_{D,IS}^{(1)}$ does not depend on the

$\gamma_{PR}^{D,IS}$ angle, since these recoupling schemes are γ-encoded.(Pileio et al., 2007; Martineau et al., 2012) The Hamiltonian of Eq.2 does not commute among different spin-pairs, and therefore the PRESTO sequence is affected by dipolar truncation, *i.e.*, the transfer to distant nuclei is attenuated by the stronger couplings with nearby spins.

As mentioned above, the SQ hetero-nuclear dipolar recoupling schemes also reintroduce the ${}^1$H CSA with the same scaling factor $\kappa$, but without commuting with the recoupled ${}^1$H-$S$ dipolar interactions. Therefore, in the case of large ${}^1$H CSA, for

instance at high magnetic fields, this interaction can interfere with the ${}^1$H-$S$ dipolar couplings, especially with small ones. These interferences can be limited by the use of the PRESTO-III variant, depicted in Fig.1a,c,(Zhao et al., 2004) in which three CT-selective pulses are applied to the $S$ nucleus. Indeed, the CT-selective π-pulses partly refocus the ${}^1$H CSA, which limits these interferences.

**II-1-2. Selection of the recoupling sequence**

On the basis of the AH and numerical simulations of spin dynamics, $R18_1^7$ and $R18_2^5$ schemes built from single rectangular π-pulses were selected for hetero-nuclear dipolar recoupling at moderate MAS frequencies, $\nu_R \approx 10$ kHz,(Zhao et al., 2001) while more recently, sequences based on symmetries $R12_5^4$, $R14_6^5$, $R16_7^6$, $R14_8^5$, $R18_8^7$, $R16_9^6$, $R20_9^8$ and $R18_{10}^7$ using $270_0 90_{180}$ as inversion element were chosen for the measurement of ${}^1$H CSA at fast MAS frequencies, $\nu_R \approx 60\text{-}70$ kHz.(Pandey et al., 2015)

We also transferred the proton polarization to ${}^{27}$Al nuclei at $\nu_R = 62.5$ kHz using PRESTO with $R16_3^2$ recoupling built from a single rectangular π-pulse.(Giovine et al., 2019a)

We screened here the $RN_n^\nu$ schemes built from single rectangular and composite π-pulses to achieve γ-encoded $|m| = 2$ SQ hetero-nuclear dipolar recoupling at $\nu_R = 20$ or $62.5$ kHz. Dipolar recoupling at $\nu_R \geq 60$ kHz is useful to correlate the signals





of quadrupolar nuclei with high-resolution $^1$H spectra without using homo-nuclear dipolar decoupling. We tested the following
composite π-pulses: $270_0 90_0 180$ with $\xi^{\text{tot}}/(2\pi) = 2$, which is offset-compensated and amplitude modulated and has been employed
in several $RN_n^\nu$ sequences,(Giovine et al., 2019a; Carravetta et al., 2000; Levitt, 2002; Pandey et al., 2015) $90_0 240_{90} 90_0$ with
$\xi^{\text{tot}}/(2\pi) = 7/3$, which compensates both rf inhomogeneity and offset,(Freeman et al., 1980; Duong et al., 2019) and
$90_{-45} 90_{45} 90_{-45}$ with $\xi^{\text{tot}}/(2\pi) = 3/2$, which has homo-nuclear decoupling properties.(Madhu et al., 2001) Adiabatic pulses cannot
be employed for SQ hetero-nuclear dipolar recoupling since they yield vanishing scaling factors for the rotational components
with μ ≠ 0.(Nagashima et al., 2018)

A total of 109 $RN_n^\nu$ symmetries with $2 \leq N \leq 30$, $2 \leq n \leq 7$ and $1 \leq \nu \leq 11$ were found which recouple {2, ±2, 1, ±1} or {2, ∓2,
1, ±1} rotational components of the $^1$H-$S$ dipolar coupling and $^1$H CSA. We selected the $RN_n^\nu$ recouplings based on those
symmetries with rf-field limited to $\nu_1 \leq 120$ and 190 for $\nu_R = 20$ and 62.5 kHz, respectively. As the currently employed γ-
encoded $|m| = 2$ SQ hetero-nuclear dipolar recoupling schemes have $50° \leq \phi \leq 70°$, we only considered $RN_n^\nu$ symmetries with
$45° \leq \phi \leq 135°$. The scaling factor, $\kappa$, of the recoupled $^1$H-$S$ dipolar interaction was calculated using the 'C and R symmetries'
Mathematica package.(Carravetta et al., 2000; Brinkmann and Levitt, 2001; Brinkmann et al., 2000; Brinkmann and Edén,
2004)

These $RN_n^\nu$ symmetries eliminate the contribution of $^1$H-$^1$H dipolar interactions to the first-order Hamiltonian, but not their
contribution to the second-order. The cross-terms between $^1$H-$^1$H interactions in the second-order Hamiltonian can be
written(Brinkmann and Edén, 2004)

$$\bar{H}^{(2),DD_1 \times DD_2} = \frac{1}{\nu_R} \sum_{\{1,2\}} \kappa_{\{1,2\}}^{DD_1 \times DD_2} \left[A_{l_2 m_2}^{DD_2}\right]^R \left[A_{l_1 m_1}^{DD_1}\right]^R \exp[i(m_1 + m_2)\omega_R t^0] \left[T_{\lambda_2 \mu_2}^{DD_2}, T_{\lambda_1 \mu_1}^{DD_1}\right] \tag{5}$$

where the sum is taken over all second-order cross-terms {**1,2**} between the {$l_1$, $m_1$, $\lambda_1$, $\mu_1$} and {$l_2$, $m_2$, $\lambda_2$, $\mu_2$} rotational
components of DD$_1$ and DD$_2$ $^1$H-$^1$H dipolar interactions, respectively.

$\kappa_{\{1,2\}}^{DD_1 \times DD_2}$ is the scaling factor of this cross-term, $\left[A_{l_i m_i}^{DD_i}\right]^R$ and $T_{\lambda_i \mu_i}^{DD_i}$ denote the component $m_i$ of the $l_i^{\text{th}}$ rank spatial irreducible
spherical tensor $A^{DD_i}$ in the MAS rotor-fixed frame and the component $\mu_i$ of the $\lambda_i^{\text{th}}$ rank spin irreducible spherical tensor
operator $T^{DD_i}$. Eq.5 indicates that the amplitude of the second-order Hamiltonian decreases at higher MAS frequency. The
magnitude of the cross-terms between $^1$H-$^1$H interactions was estimated by calculating the Euclidean norm(Hu et al., 2009;
Gansmüller et al., 2013)

$$\left\| \kappa_{\{1,2\}}^{DD_1 \times DD_2} \right\|_2 = \sqrt{\sum_{\{1,2\}} \left| \kappa_{\{1,2\}}^{DD_1 \times DD_2} \right|^2}. \tag{6}$$

For each basic element $\mathcal{R}$, we selected the $RN_n^\nu$ schemes with the highest ratio $\kappa / \left\| \kappa_{\{1,2\}}^{DD_1 \times DD_2} \right\|_2$ in order to minimize the
interference of $^1$H-$^1$H dipolar interactions with the $^1$H-$S$ dipolar recoupling. Besides $^1$H-$^1$H dipolar interactions, other cross-
terms involving $^1$H CSA and offset can also interfere with the $^1$H-$S$ dipolar recoupling. These cross-terms can be expressed by
Eq.5, in which DD$_1$ and DD$_2$ indexes are substituted by other interactions, such as $^1$H CSA or isotropic chemical shift ($\delta_{\text{iso}}$).



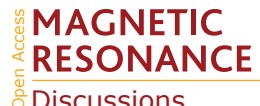

For the selected symmetries, we estimated the magnitude of the cross-terms between $^1$H CSA or offset by calculating the Euclidean norms $\left\|\kappa_{\{1,2\}}^{CSA \times CSA}\right\|_2$ and $\left\|\kappa_{\{1,2\}}^{\delta iso \times \delta iso}\right\|_2$ given by Eq.6.

The corresponding selected $RN_n^\nu$ sequences are listed in Table 1 for $\nu_R$ = 20 kHz and Table 2 for $\nu_R$ = 62.5 kHz.

For $\nu_R$ = 20 kHz, according to the AH, the $RN_n^\nu$ sequence with the highest robustness to $^1$H-$^1$H dipolar interactions is $R22_2^7(180_0)$. However, this recoupling is slightly less robust to $^1$H CSA and offset than $R18_2^5(180_0)$, which has already been reported. For this MAS frequency, the $RN_n^\nu$ schemes using the chosen composite pulses either required rf-fields greater than 120 kHz, e.g. $\nu_1$ = 130 and 173 kHz for the $R26_3^7$ schemes built from $90_{-45}90_{45}90_{-45}$ and $270_090_{180}$ pulses, or did not suppress efficiently the second-order cross-terms between $^1$H-$^1$H interactions because of small rf-field ($\nu_1 \leq 62.5$ kHz).

**Table 1. Selected $RN_n^\nu$ $|m|$ = 2 SQ hetero-nuclear dipolar recoupling for $\nu_R$ = 20 kHz.**

| $\mathcal{R}$ | $RN_n^\nu$ | $\phi°$ | $\nu_1/\nu_R$ | $\kappa$ | $\kappa/\left\|\kappa_{\{1,2\}}^{DD_1 \times DD_2}\right\|_2$ | $\kappa/\left\|\kappa_{\{1,2\}}^{CSA \times CSA}\right\|_2$ | $\kappa/\left\|\kappa_{\{1,2\}}^{\delta iso \times \delta iso}\right\|_2$ |
|---|---|---|---|---|---|---|---|
| | $R22_2^7$ | 57 | 5.5 | 0.178 | 162 | 7.12 | 17.58 |
| $180_0$ | $R28_3^5$ | 51 | 4.67 | 0.176 | 156 | 5.08 | 18.29 |
| | $R18_2^5$ | 50 | 4.5 | 0.175 | 140 | 7.20 | 18.49 |

For $\nu_R$ = 62.5 kHz, the $RN_n^\nu$ sequences using composite $\pi$-pulses recouple the $^1$H-$S$ dipolar interaction with a higher scaling factor than those built from single $\pi$-pulses. According to AH, the $90_0240_{90}90_0$ basic element leads to the highest robustness to $^1$H-$^1$H interferences. Even if the amplitude of the cross-terms is inversely proportional to the MAS frequency (Eq.5), the amplitude of these terms is lower at $\nu_R$ = 20 than 62.5 kHz. The $270_090_{180}$ element is less robust to $^1$H-$^1$H interferences, but benefits from a high robustness to offset. The selected $RN_n^\nu$ symmetries for this element include $R14_6^5$ and $R16_7^6$, which have already been employed for the measurement of $^1$H CSA and the transfer of $^1$H polarization to half-integer quadrupolar nuclei at $\nu_R \geq 60$ kHz.(Giovine et al., 2019a; Pandey et al., 2015) As the scaling factors $\kappa$ of the $^1$H-$S$ dipolar interaction of the $RN_n^\nu$ schemes built from single $\pi$-pulses with $45° \leq \phi \leq 135°$ are small, we also selected in Table 3 $RN_n^\nu$ schemes built from single $\pi$-pulses with $\kappa \geq 0.15$, but with extended $\phi$ ranges. These recoupling schemes are less robust to offset than the $RN_n^\nu$ schemes built from $270_090_{180}$ element.

**Table 2. Selected $RN_n^\nu$ $|m|$ = 2 SQ hetero-nuclear dipolar recoupling with $45° \leq \phi \leq 135°$ for $\nu_R$ = 62.5 kHz.**

| $\mathcal{R}$ | $RN_n^\nu$ | $\phi°$ | $\nu_1/\nu_R$ | $\kappa$ | $\kappa/\left\|\kappa_{\{1,2\}}^{DD_1 \times DD_2}\right\|_2$ | $\kappa/\left\|\kappa_{\{1,2\}}^{CSA \times CSA}\right\|_2$ | $\kappa/\left\|\kappa_{\{1,2\}}^{\delta iso \times \delta iso}\right\|_2$ |
|---|---|---|---|---|---|---|---|
| | $R10_4^3$ | 54 | 2.92 | 0.227 | 39.63 | 2.82 | 12.63 |
| $90_0240_{90}90_0$ | $R14_6^5$ | 64.3 | 2.72 | 0.232 | 36.33 | 1.87 | 12.39 |



| $\mathcal{R}$ | $RN_n^\nu$ | $\phi\,/°$ | $\nu_1/\nu_R$ | $\kappa$ | $\kappa/\left\|\kappa_{\{1,2\}}^{DD_1\times DD_2}\right\|_2$ | $\kappa/\left\|\kappa_{\{1,2\}}^{CSA\times CSA}\right\|_2$ | $\kappa/\left\|\kappa_{\{1,2\}}^{\delta iso\times\delta iso}\right\|_2$ |
|---|---|---|---|---|---|---|---|
| | $R12_5^4$ | 60 | 2.80 | 0.230 | 36.08 | 2.25 | 12.47 |
| | $R12_7^8$ | 120 | 2.00 | 0.227 | 35.96 | 1.61 | 7.72 |
| $270_090_{180}$ | $R16_7^6$ | 67.5 | 2.28 | 0.150 | 17.96 | 1.85 | $3.50\times10^{10}$ |
| | $R16_7^{10}$ | 112.5 | 2.28 | 0.150 | 17.96 | 1.85 | $3.50\times10^{10}$ |
| | $R14_6^5$ | 64.3 | 2.33 | 0.150 | 15.90 | 2.33 | $3.58\times10^{10}$ |
| | $R14_6^9$ | 115.7 | 2.33 | 0.150 | 15.90 | 2.15 | $3.58\times10^{10}$ |
| $90_{-45}90_{45}90_{-45}$ | $R10_4^3$ | 54 | 1.88 | 0.186 | 16.70 | 2.97 | 15.07 |
| | $R18_7^5$ | 50 | 1.93 | 0.189 | 15.73 | 1.98 | 25.49 |
| | $R14_6^5$ | 64.3 | 1.75 | 0.177 | 15.55 | 2.09 | 5.49 |
| | $R12_5^4$ | 60 | 1.80 | 0.181 | 15.17 | 2.47 | 8.11 |
| $180_0$ | $R14_6^5$ | 64.3 | 1.16 | 0.085 | 5.35 | 2.26 | 1.34 |
| | $R14_6^9$ | 115.7 | 1.16 | 0.085 | 5.35 | 2.26 | 1.34 |
| | $R16_7^6$ | 67.5 | 1.14 | 0.082 | 4.90 | 1.98 | 1.09 |
| | $R16_7^{10}$ | 112.5 | 1.14 | 0.082 | 4.90 | 1.98 | 1.09 |

**Table 3. Selected $RN_n^\nu$ $|m|$ = 2 SQ hetero-nuclear dipolar recoupling built from single π pulses with 20° ≤ ϕ ≤ 160° and κ ≥ 0.15 for $\nu_R$ = 62.5 kHz.**

| $\mathcal{R}$ | $RN_n^\nu$ | $\phi\,/°$ | $\nu_1/\nu_R$ | $\kappa$ | $\kappa/\left\|\kappa_{\{1,2\}}^{DD_1\times DD_2}\right\|_2$ | $\kappa/\left\|\kappa_{\{1,2\}}^{CSA\times CSA}\right\|_2$ | $\kappa/\left\|\kappa_{\{1,2\}}^{\delta iso\times\delta iso}\right\|_2$ |
|---|---|---|---|---|---|---|---|
| | $R28_5^4$ | 25.7 | 2.75 | 0.163 | 24.42 | 3.34 | 26.42 |
| $180_0$ | $R22_4^3$ | 24.5 | 2.75 | 0.162 | 22.84 | 4.10 | 27.24 |
| | $R16_3^2$ | 22.5 | 2.67 | 0.161 | 16.26 | 5.21 | 28.89 |

## II-2. *D*-RINEPT

### II-2-1. Zero-quantum hetero-nuclear dipolar recoupling

In the *D*-RINEPT sequence, the $^1$H-*S* dipolar interactions are reintroduced under MAS by applying non-γ-encoded two-spin order dipolar recoupling to the $^1$H channel. These recoupling schemes reintroduce the $|m|$ = 2 space components and the zero-quantum (0Q) terms of the $^1$H-*S* dipolar interaction and $^1$H CSA, *i.e.,* the rotational components {$l, m, \lambda, \mu$} = {2, ±2, 1, 0}, while they suppress the contributions of $^1$H isotropic chemical shifts, the hetero-nuclear *J*-couplings with protons, and the $^1$H-$^1$H dipolar couplings to the first-order average Hamiltonian.(Brinkmann and Kentgens, 2006a, b) The contribution of the $^1$H-*S* dipolar coupling to this Hamiltonian is equal to:(Giovine et al., 2019a; Brinkmann and Kentgens, 2006a; Lu et al., 2012)





$$\overline{H}_{D,IS}^{(1)} = 2\omega_{D,IS}I_z S_z \tag{7}$$

where

$$\omega_{D,IS} = \kappa b_{IS}\sin^2\left(\beta_{PR}^{D,IS}\right)\cos(2\varphi), \tag{8}$$

The norm of $\overline{H}_{D,IS}^{(1)}$ depends on the $\varphi$ phase, given by Eq.4, and hence on the $\gamma_{PR}^{D,IS}$ angle. Therefore, these two-spin order dipolar

recoupling schemes are non-$\gamma$-encoded. The Hamiltonian of Eq.7 commutes among different spin pairs and hence, these recoupling schemes are not affected by dipolar truncation. Similarly, the recoupled $^1$H CSA contribution to the first-order Hamiltonian is proportional to $I_z$ and hence, commutes with the recoupled $^1$H-$S$ dipolar interactions and does not interfere with the hetero-nuclear dipolar recoupling.

**II-2-2. Selection of the recoupling sequence**

Different R$N_n^\nu$ sequences have been proposed to achieve non-$\gamma$-encoded $|m| = 2$ two-spin order dipolar recoupling, including (i) symmetries R$(4n)_n^{2n-1} =$ R$12_3^5$, R$16_4^7$, R$20_5^9$, R$24_6^{11}$, R$28_7^{13}$ and R$32_8^{15}$ for $n = 3, 4, 5, 6, 7$ and 8 using single $\pi$-pulses as basic element, which have been employed to measured $^1$H-$^{17}$O dipolar couplings at $\nu_R = 50$ kHz,(Brinkmann and Kentgens, 2006b) (ii) SR$4_1^2$ recoupling built from a single $\pi$-pulse, which corresponds to the $[R4_1^2 R4_1^{-2}]_0 [R4_1^2 R4_1^{-2}]_{120} [R4_1^2 R4_1^{-2}]_{240}$ sequence and has been employed in the RINEPT scheme,(Nagashima et al., n.d.; Giovine et al., 2019a) (iii) R$12_3^5$ and SR$4_1^2$

schemes using a $90_{-45}90_{45}90_{-45}$ composite $\pi$-pulse as a basic element, which have been incorporated into $D$-HMQC sequences at $\nu_R = 36$ kHz,(Perras et al., 2019) and (iv) SR$4_1^2$ schemes built from a tanh/tan adiabatic pulse, which have been used in RINEPT sequence at $\nu_R \approx 36$ kHz.(Nagashima et al., 2020, n.d.) During the tanh/tan pulse, the instantaneous rf-amplitude is equal to:

$$\omega_1(t) = \omega_{1,max} \begin{cases} \tanh\left[\frac{8\xi t}{T_R}\right] & 0 \le t < T_R/8 \\ \tanh\left[2\xi\left(1 - \frac{4t}{T_R}\right)\right] & T_R/8 \le t < T_R/4 \end{cases} \tag{9}$$

where $\omega_{1,max}$ is the peak amplitude of the rf-field, $t$ refers to the time since the start of the pulse, which lasts $T_R/4$ when incorporated into the SR$4_1^2$ recoupling scheme. The parameter $\xi$ determines the rise and fall times of the pulse. Hence, in the frequency-modulated (FM) frame,(Garwood and DelaBarre, 2001) the frequency offset i

$$\phi_I(t) = \frac{\Delta\nu_{0,max}}{2\theta\tan(\theta)}\ln\left\{\cos\left[\theta\left(1 - 8\frac{t}{T_R}\right)\right]\right\}. \tag{10}$$

where $\Delta\nu_{0,max}$ is the peak amplitude of the carrier frequency modulation and $\theta$ determines the frequency sweep rate in the

center of the pulse. Here, we employed $\xi = 10$ and $\theta = 87° =$ atan(20).(Nagashima et al., 2020; Kervern et al., 2007; Nagashima et al., 2018)

We screened here the R$N_n^\nu$ schemes built from $180_0$, $270_0 90_{180}$, $90_0 240_{90} 90_0$ and $90_{-45} 90_{45} 90_{-45}$ elements. A total of 58 R$N_n^\nu$ symmetries with $2 \le N \le 30$, $2 \le n \le 7$ and $1 \le \nu \le 11$ were found which recouple the $\{2, \pm 2, 1, 0\}$ rotational components of



the $^1$H-$S$ dipolar coupling and $^1$H CSA. We only considered the R$N_n^\nu$ symmetries with $60° \leq \phi \leq 120°$ since the currently

employed non-γ-encoded $|m| = 2$ two-spin order hetero-nuclear dipolar recoupling schemes have $75° \leq \phi \leq 90°$.

We calculated the scaling factor of the recoupled $^1$H-$S$ dipolar interaction and the Euclidean norm and $\left\| \kappa_{\{1,2\}}^{DD_1 \times DD_2} \right\|_2$ of the

cross-terms between $^1$H-$^1$H interactions using the 'C and R symmetries' Mathematica package.(Carravetta et al., 2000;

Brinkmann and Levitt, 2001; Brinkmann et al., 2000; Brinkmann and Edén, 2004) For each basic element $\mathcal{R}$, we selected the

R$N_n^\nu$ schemes with the highest ratios $\kappa / \left\| \kappa_{\{1,2\}}^{DD_1 \times DD_2} \right\|_2$. The selected R$N_n^\nu$ sequences are listed in Table 4. The parameters of the

SR$4_1^2$ schemes built from the different basic element $\mathcal{R}$ are also listed in Table 4 for the sake comparison. For those R$N_n^\nu$

sequences, we calculated the Euclidean norms $\left\| \kappa_{\{1,2\}}^{CSA \times CSA} \right\|_2$ and $\left\| \kappa_{\{1,2\}}^{\delta iso \times \delta iso} \right\|_2$ in order to estimate the magnitudes of the cross-

terms between $^1$H CSA and offset.

According to the AH, the $90_0240_{90}90_0$ composite π-pulse yields the highest robustness to $^1$H-$^1$H dipolar interactions. However,

the rf-field requirement of the R$N_n^\nu$ sequences built from this composite pulse, $\nu_1 = 4.66\nu_R$, $i.e.$, $\nu_1 = 291$ kHz at $\nu_R = 62.5$ kHz,

is not compatible with most 1.3 mm MAS probes. Furthermore, the highest robustness to $^1$H CSA and offset is achieved using

the $270_090_{180}$ composite π-pulse. The SR$4_1^2$ schemes benefit from the highest robustness to $^1$H CSA, because of the three-step

multiple-quantum super-cycle.(Brinkmann and Edén, 2004; Brinkmann and Kentgens, 2006a) Contrary to the R$N_n^\nu$ $|m| = 2$

SQ hetero-nuclear dipolar recouplings, the rf-field of the R$N_n^\nu$ $|m| = 2$ two-spin orders is always higher than $2\nu_R$ since these

R$N_n^\nu$ symmetries with $2n > N$, such as R$12_9^5$, lead to vanishing $\kappa$ scaling factor.

In the case of the adiabatic R$N_n^\nu$ (tt) sequences, the determination of the scaling factors of first- and second-order terms of the

effective Hamiltonian is more cumbersome since they depend on the $\nu_{1,max}$, $\Delta\nu_{0,max}$, $\xi$ and $\theta$ parameters.(Nagashima et al.,

2018) For example, the scaling factor of the R$12_3^5$ and SR$4_1^2$ schemes is $\kappa = 0.31$ for $\nu_{1,max}/\Delta\nu_{0,max} = 0.685$, $\xi = 10$ and $\theta = 87°$,

and this value monotonously decreases for increasing $\nu_{1,max}/\Delta\nu_{0,max}$ ratios.

**Table 4. Selected R$N_n^\nu$ $|m| = 2$ two-spin order hetero-nuclear dipolar recoupling.**

| $\mathcal{R}$ | R$N_n^\nu$ | $\phi°$ | $\nu_1/\nu_R$ | $\kappa$ | $\kappa / \left\| \kappa_{\{1,2\}}^{DD_1 \times DD_2} \right\|_2$ | $\kappa / \left\| \kappa_{\{1,2\}}^{CSA \times CSA} \right\|_2$ | $\kappa / \left\| \kappa_{\{1,2\}}^{\delta iso \times \delta iso} \right\|_2$ |
|---|---|---|---|---|---|---|---|
| | R$16_4^9$ | 101 | 4.66 | 0.131 | 63.17 | 16.48 | 9.31 |
| | R$20_5^{11}$ | 99 | 4.66 | 0.131 | 60.68 | 16.59 | 14.45 |
| | R$12_3^7$ | 105 | 4.66 | 0.131 | 51.25 | 16.11 | 9.70 |
| $90_0240_{90}90_0$ | R$16_4^7$ | 79 | 4.66 | 0.131 | 45.52 | 15.76 | 13.60 |
| | R$28_7^{10}$ | 64 | 4.66 | 0.131 | 44.55 | 14.06 | 11.98 |
| | R$20_5^9$ | 81 | 4.66 | 0.131 | 44.30 | 15.95 | 14.46 |
| | R$12_3^5$ | 75 | 4.66 | 0.131 | 43.91 | 15.40 | 12.83 |



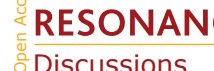

| | | | | | | |
|---|---|---|---|---|---|---|
| | $SR4_1^2$ | 90 | 4.66 | 0.131 | 42.37 | 22.65 | 10.48 |
| | $R28_7^{11}$ | 71 | 3 | 0.191 | 39.81 | 10.05 | 6.10 |
| | $R20_5^8$ | 72 | 3 | 0.191 | 39.74 | 10.26 | 5.49 |
| | $R8_2^3$ | 67.5 | 3 | 0.191 | 39.43 | 9.42 | 7.88 |
| $90_{-45}90_{45}90_{-45}$ | $R8_2^{11}$ | 67.5 | 3 | 0.191 | 39.43 | 9.42 | 7.88 |
| | $R24_6^{10}$ | 75 | 3 | 0.191 | 39.32 | 10.66 | 4.22 |
| | $R28_7^{10}$ | 64.3 | 3 | 0.191 | 38.82 | 8.65 | 10.13 |
| | $R12_3^5$ | 75 | 3 | 0.191 | 38.33 | 10.66 | 4.22 |
| | $SR4_1^2$ | 90 | 3 | 0.191 | 19.95 | 19.48 | 1.33 |
| | $R24_6^{11}$ | 82.5 | 4 | 0.212 | 33.12 | 25.46 | $8.67\times10^{10}$ |
| | $R20_5^9$ | 81 | 4 | 0.212 | 31.85 | 25.19 | $8.67\times10^{10}$ |
| | $R20_5^{11}$ | 99 | 4 | 0.212 | 31.85 | 25.19 | $8.67\times10^{10}$ |
| $270_090_{180}$ | $R16_4^7$ | 78.8 | 4 | 0.212 | 28.56 | 24.69 | $8.67\times10^{10}$ |
| | $R16_4^9$ | 101.2 | 4 | 0.212 | 28.56 | 24.69 | $8.67\times10^{10}$ |
| | $R12_3^5$ | 75 | 4 | 0.212 | 20.84 | 23.58 | $8.67\times10^{10}$ |
| | $R12_3^7$ | 105 | 4 | 0.212 | 20.84 | 23.58 | $8.67\times10^{10}$ |
| | $SR4_1^2$ | 90 | 4 | 0.212 | 35.21 | 149.93 | $8.67\times10^{10}$ |
| | $R16_4^7$ | 78.8 | 2 | 0.25 | 19.65 | 10.52 | 2.78 |
| | $R16_4^9$ | 115.7 | 2 | 0.25 | 19.65 | 10.52 | 2.78 |
| $180_0$ | $R12_3^5$ | 75 | 2 | 0.25 | 18.9 | 9.89 | 3.74 |
| | $R12_3^7$ | 105 | 2 | 0.25 | 18.9 | 9.89 | 3.74 |
| | $SR4_1^2$ | 90 | 2 | 0.25 | 13.2 | 22.98 | 1.56 |

**II-2-3. *D*-RINEPT-CWc sequence**

The *D*-RINEPT-CWc sequence is displayed in Fig.1b,c. The $^1$H-$S$ dipolar couplings are reintroduced by applying the $RN_n^\nu$ schemes listed in Table 4 during the defocusing and refocusing delays τ, which are identical in this article, even if distinct defocusing and refocusing delays can improve the transfer efficiency.(Nagashima et al., 2020) As the two-spin order recoupling schemes are non-γ-encoded, they must be rotor-synchronized. We used here a delay of $T_R$ between two successive $RN_n^\nu$ blocks. In the *D*-RINEPT-CWc sequence, a CW irradiation is applied during these delays in order to limit the losses due to $^1$H-$^1$H dipolar interactions.(Nagashima et al., n.d.) The nutation during this CW irradiation is eliminated by employing CW irradiations with opposite phases. Furthermore, the robustness to $^1$H rf-field inhomogeneity is improved by replacing the first



π and second π/2 pulses by composite $90_0180_{90}90_0$ and $90_{90}90_0$ pulses, respectively, the CW irradiation being applied between the individual pulses.(Freeman et al., 1980; Levitt and Freeman, 1979)

## III. Numerical simulations

### III-1. Simulation parameters

All simulations were performed using the version 4.1.1 of SIMPSON package (Bak et al., 2000, p.200). The powder average
calculation was performed using 462 $\{\alpha_{MR}, \beta_{MR}, \gamma_{MR}\}$ Euler angles relating the rotor and molecular frames. This set of angles was obtained by considering 66 $\{\alpha_{MR}, \beta_{MR}\}$ pairs and 7 $\gamma_{MR}$ angles. The $\{\alpha_{MR}, \beta_{MR}\}$ values were selected according to the REPULSION algorithm (Bak and Nielsen, 1997b), while the $\gamma_{MR}$ angles were regularly stepped from 0 to 360°.

To accelerate the simulations, the $^1H \to {}^{15}N$ RINEPT transfer was used, instead of the $^1H \to {}^{27}Al$ one, because the computing time is proportional to the cube of the size of the density matrix. Furthermore, in RINEPT experiments, only CT-selective
pulses are applied to the quadrupolar nuclei and hence, the contribution of STs to the signal can be disregarded. The $^1H \to {}^{15}N$ RINEPT transfer was simulated for a $^{15}N^1H_4$ spin system. A similar approach has already been applied for the simulation of the RINEPT transfer from protons to quadrupolar nuclei (Nagashima et al., n.d.; Giovine et al., 2019b). This $^{15}N^1H_4$ spin system comprises a tetrahedron of four protons with a $^{15}N$ nucleus on one of its symmetry axis. The dipolar coupling constants between protons are all equal to $|b_{HH}|/(2\pi) = 1, 7$ or 15 kHz. The dipolar coupling between $^{15}N$ nucleus and its closest $^1H$
neighbor is $|b_{HN}|/(2\pi) = 2575$ Hz, corresponding to a $^1H$-$^{27}Al$ distance of 2.3 Å, typical of the distance between the protons of hydroxyl groups and the Al atoms of the first surface layer of hydrated γ-alumina (Lee et al., 2014). All protons were subject to a CSA of 6 kHz, *i.e.*, 7.5 ppm at 18.8 T, their asymmetry parameters were null, and their principal axis coincide with the 3-fold rotational axes of the $^1H_4$ tetrahedron.

The simulations were performed for a static magnetic field of 18.8 T, for which the $^1H$ and $^{15}N$ Larmor frequencies were equal
to 800 and 81 MHz, respectively using MAS frequencies of $\nu_R = 20$ or 62.5 kHz (Liang et al., 2018). $^1H \to {}^{15}N$ RINEPT-CWc sequences incorporating either $SR4_1^2$(tt) or $R12_3^5$(tt) recoupling schemes were simulated. The defocusing and refocusing periods were both equal to their optimal values τ = 650 or 640 μs at $\nu_R = 20$ or 62.5 kHz, respectively. The rf-field nutation frequency on the $^1H$ channel was equal to 200 kHz during the π/2 and π-pulses, which do not belong to the recoupling sequence, as well as the CW irradiation, whereas the pulses applied to $S = {}^{15}N$ nuclei were considered as ideal Dirac pulses. Simulations
were performed for recoupling schemes made of tanh/tan adiabatic pulses with $\nu_{1,max}$ and $\Delta\nu_{0,max}$ parameters ranging from $0.5\nu_R$ to $10\nu_R$ and from $10\nu_R$ to $200\nu_R$, respectively. The other pulses were applied on resonance. The density matrix before the first pulse was equal to $I_{1z} + I_{2z} + I_{3z} + I_{4z}$. We normalized the transfer efficiency of $^1H \to {}^{15}N$ RINEPT sequences to the maximal signal for a $^1H \to {}^{15}N$ through-bond RINEPT sequence made of ideal Dirac pulses in the case of a $^{15}N$-$^1H$ spin system with a *J*-coupling constant of 150 Hz.




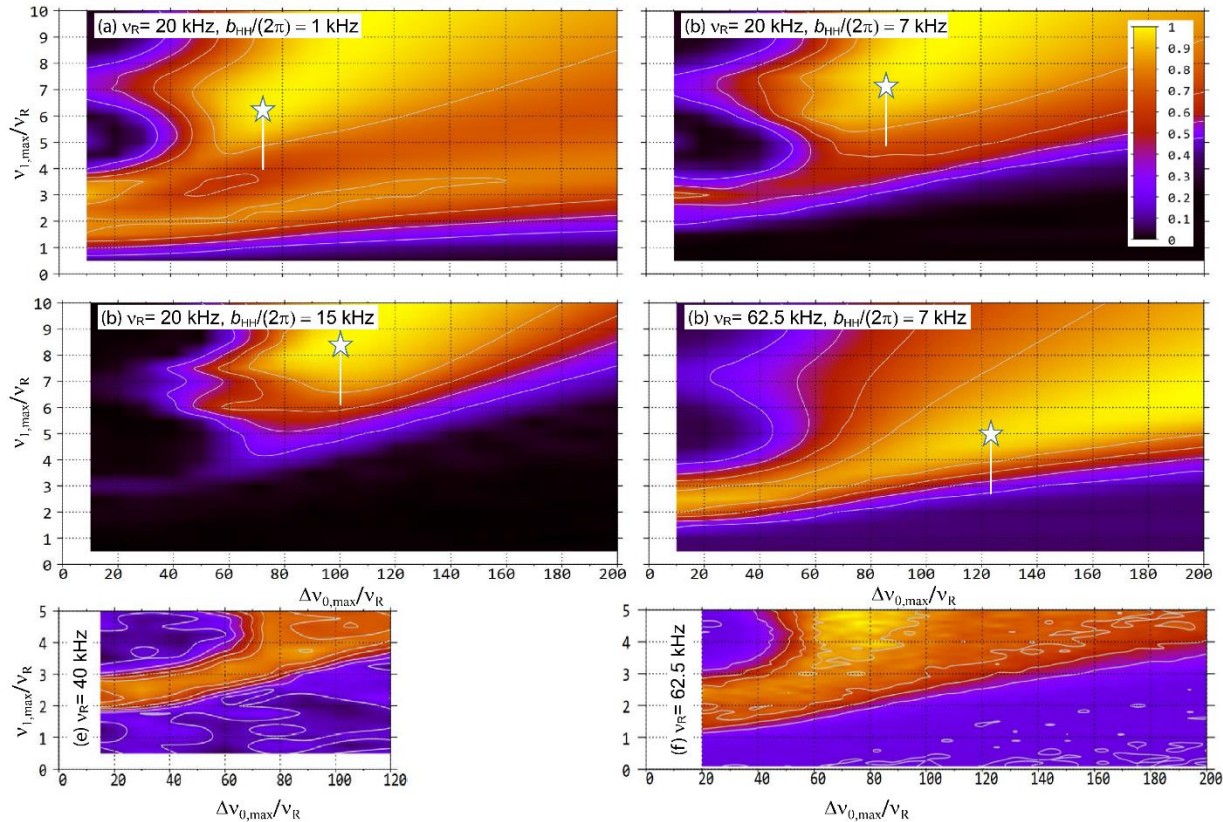

**Figure 2: (a-d)** Simulated transfer efficiency of $^1H \rightarrow {}^{15}N$ $D$-RINEPT-SR4$_1^2$(tt) sequence for a $^{15}N^1H_4$ spin system as function of $\nu_{1,max}/\nu_R$ and $\Delta\nu_{0,max}/\nu_R$ for $\nu_R = 20$ and 62.5 kHz and $b_{HH}/(2\pi) =$ **(a)** 1, **(b,d)** 7 and **(c)** 15 kHz. **(e,f)** Experimental $^1H \rightarrow {}^{15}N$ $D$-RINEPT-SR4$_1^2$(tt) signal of L-histidine·HCl as function of $\nu_{1,max}/\nu_R$ and $\Delta\nu_{0,max}/\nu_R$ at 18.8 T with $\nu_R =$ **(e)** 40 (e) or **(f)** 62.5 kHz. The white star indicates recoupling conditions with minimal rf field leading to maximal transfer efficiency. The white vertical line mimics the rf-field distribution within the coil.

## III-2. Optimal adiabatic recoupling

The transfer efficiency of RINEPT using R$N_n^\nu$ schemes built from adiabatic pulses, depends on $\nu_{1,max}$ and $\Delta\nu_{0,max}$ parameters. For a similar $^{15}N^1H_4$ spin system with $|b_{HN}|/(2\pi) = 2.575$ and $|b_{HH}|/(2\pi) = 7$ kHz, spinning at $\nu_R = 12.5$ kHz, we showed using numerical simulations of spin dynamics that a maximal transfer efficiency was achieved provided that $\nu_{1,max} = 0.07\Delta\nu_{0,max}$ and $\nu_{1,max}/\nu_R \geq 8$.(Nagashima et al., n.d.) In practice, we used $\nu_{1,max} = 11\nu_R = 137$ kHz and $\Delta\nu_{0,max} = 160\nu_R = 2$ MHz.

Similar simulations were performed here for $\nu_R = 20$ or 62.5 kHz. As seen in Fig.2a-c, at a given MAS frequency, higher $^1H$-$^1H$ dipolar couplings require higher rf-field and broader carrier frequency sweep so that the tanh/tan pulses remain adiabatic in spite of the modulation of the $^1H$-$^1H$ dipolar couplings by MAS.(Nagashima et al., n.d.; Kervern et al., 2007) For $|b_{HH}|/(2\pi)$ = 7 kHz, the minimal $\nu_{1,max}/\nu_R$ ratio decreases for higher MAS frequencies (compare Figs.2b and d) since the contribution of





the modulation of $^1$H-$^1$H dipolar couplings by MAS to the first adiabaticity factor is proportional to $(\nu_{1,max})^2/\nu_R$ and hence, $\nu_{1,max}$ values proportional to $\sqrt{\nu_R}$, i.e. $\nu_{1,max}/\nu_R$ ratio inversely proportional to $\sqrt{\nu_R}$, are sufficient to maintain the adiabaticity of the pulses.(Kervern et al., 2007) Nevertheless, Fig.2d indicates that SR4$_1^2$(tt) recoupling requires $\nu_{1,max} \geq 313$ kHz for $\nu_R = 62.5$

kHz. This rf field is not compatible with the specifications of most 1.3 mm MAS probes. Similar transfer efficiencies were simulated for the RINEPT sequence with R12$_3^5$(tt) recoupling scheme (not shown).

## IV. NMR experiments

### IV-1. Samples and experimental conditions

L-[U-$^{15}$N]-histidine·HCl (hereafter referred to as "histidine") and isotopically unmodified γ-alumina were purchased from
Merck, and AlPO$_4$-14 was prepared as described previously.(Antonijevic et al., 2006)

All $^1$H → S RINEPT-CWc and PRESTO-III experiments were performed at B$_0$ = 18.8 T on Bruker BioSpin Avance NEO spectrometers equipped with double-resonance $^1$H/X probes.

$^1$H→$^{15}$N RINEPT-CWc experiments using SR4$_1^2$(tt) recoupling (denoted RINEPT-CWc-SR4$_1^2$(tt) hereafter) on histidine were performed with 1.3 and 0.7 mm MAS probes spinning at $\nu_R = 40$ or 62.5 kHz, with defocusing and refocusing delays equal to
$\tau = 375$ or 384 µs, respectively. The rf-field of the $^1$H $\pi/2$ and $\pi$ pulses, which do not belong to the recoupling scheme, was equal to 200 kHz, that of the continuous wave irradiation to $\nu_{1,CW} = 100$ kHz, and that of the $^{15}$N pulses to 62 kHz. $^1$H decoupling with a rf-field of 16 kHz was applied during the acquisition. The pulses on the $^1$H channel were applied on resonance, whereas those on $^{15}$N channel were applied at the isotropic chemical shift of the $^{15}$NH$^\tau$ signal (172 ppm). These 1D spectra resulted from averaging 8 transients with a relaxation delay of 3 s. The $^{15}$N isotropic chemical shifts were referenced to an aqueous
saturated solution of NH$_4$NO$_3$ using [$^{15}$N]-glycine as a secondary reference.

$^1$H→$^{27}$Al RINEPT-CWc and PRESTO-III experiments on γ-alumina and AlPO$_4$-14 were performed with a 1.3 mm MAS probe spinning at $\nu_R = 20$ (to test the R$N_n^\nu$ schemes with large rf-fields) or 62.5 kHz. The tested recoupling schemes are listed in Tables 5 and 6 for $\nu_R = 20$ kHz and Tables 7 and 8 for $\nu_R = 62.5$ kHz. The rf-field of the $^1$H $\pi/2$ and $\pi$ pulses, which do not belong to the recoupling scheme, was equal to 208 kHz, that of the continuous wave irradiation to $\nu_{1,CW} = 147$ kHz, and the
$^{27}$Al CT-selective one for $\pi/2$ and $\pi$ pulses to 10 kHz. The defocusing and refocusing delays $\tau$ are given in Table 5 to 8. The pulses on the $^1$H channel were applied on resonance, whereas those on $^{27}$Al channel were applied (i) on resonance with AlO$_6$ signal of γ-alumina in Figs.4 and 7, Tables 5 and 7, as well as in Figs. 5 and 8 when the offset is null, (ii) on resonance with AlO$_4$ signal of AlPO$_4$-14 in Figs.S1 and S3, Tables 6 and 8 as well as in Figs.S2 and S4 when the offset is null, and (iii) in the middle of the AlO$_4$ and AlO$_6$ peaks for the 1D spectra shown in Figs.3 and 6. These differences in offset explain some changes
in the relative efficiencies of the recoupling between the figures. These 1D spectra resulted from averaging 64 transients with a relaxation delay of 1 s The $^{27}$Al isotropic chemical shifts were referenced at 0 ppm to 1 mol.L$^{-1}$ [Al(H$_2$O)$_6$]$^{3+}$ solution.



MAGNETIC RESONANCE
Discussions
## IV-2. Optimal adiabatic recoupling

Figs.**2e** and **f** show the efficiency of $^1H \rightarrow ^{15}N$ RINEPT-SR4$_1^2$(tt) transfer for histidine as function of $\nu_{1,max}/\nu_R$ and $\Delta\nu_{0,max}/\nu_R$

for $\nu_R$ = 40 or 62.5 kHz, respectively. These experimental data indicate that at higher MAS frequency, an efficient adiabatic

recoupling can be achieved for lower $\nu_{1,max}/\nu_R$ and $\Delta\nu_{0,max}/\nu_R$ ratios. This result agrees with the numerical simulations of Figs.2b

and d.

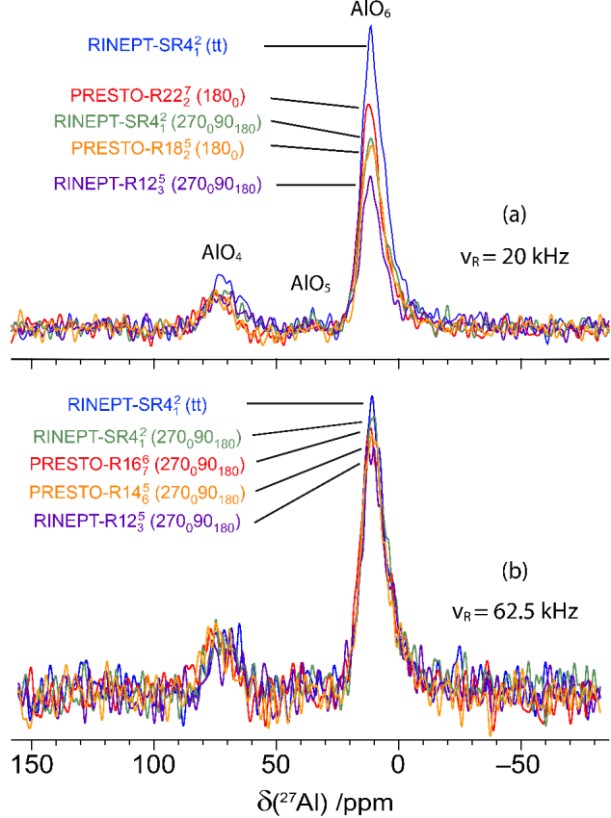

**Figure 3: 1D $^{27}Al$ spectra of γ-alumina at B$_0$ = 18.8 T with $\nu_R$ = (a) 20 and (b) 62.5 kHz acquired using $^1H \rightarrow ^{27}Al$ RINEPT-CWc and PRESTO-III transfers using the following recoupling schemes: SR4$_1^2$(tt), SR4$_1^2$(270$_0$90$_{180}$) or R12$_3^5$ (270$_0$90$_{180}$) for RINEPT and**

**R22$_2^7$(180$_0$) or R18$_2^5$ (180$_0$) for PRESTO at $\nu_R$ = 20 kHz, and R16$_7^6$ (270$_0$90$_{180}$) or R14$_6^5$ (270$_0$90$_{180}$) for PRESTO at $\nu_R$ = 62.5 kHz (b). τ delay and $\nu_1/\nu_{1,max}$ rf-field were fixed to their optimum values given in Tables 5 and 7.**

### IV-3. PRESTO and RINEPT performances for $\nu_R$ = 20 kHz

### IV-3-1. γ-alumina

The 1D NMR spectra of γ-alumina acquired using $^1H \rightarrow ^{27}Al$ RINEPT and PRESTO sequences, shown in Fig.3, exhibit two

resonances at 70 and 10 ppm, assigned to tetra- (AlO$_4$) and hexa-coordinated (AlO$_6$) resonances, respectively.(Morris and



Ellis, 1989) The signal of penta-coordinated (AlO$_5$) sites, which are mainly located in the first surface layer, is barely detected because of the lack of sensitivity of conventional solid-state NMR spectroscopy.(Lee et al., 2014, p.201) The most intense peak, AlO$_6$, was used to compare the transfer efficiencies of RINEPT and PRESTO sequences with different recoupling

schemes.

Table 5 lists the measured performances of $^1$H → $^{27}$Al RINEPT-CWc and PRESTO transfers using various recoupling for γ-alumina at $\nu_R = 20$ kHz. We notably compared the PRESTO sequences using $R22_2^7(180_0)$ and $R18_2^5(180_0)$ recoupling (Table 1) with the RINEPT-CWc scheme using a recoupling based on $SR4_1^2$ and $R12_3^5$ symmetries with: single $180_0$, composite $270_090_{180}$ and $90_{-45}90_{45}90_{-45}$ or tanh/tan adiabatic pulses. A low transfer efficiency was obtained for RINEPT-CWc-

$SR4_1^2(90_0240_{90}90_0)$ because of its low scaling factor κ = 0.131 and hence, its performances are not reported in Table 5. We also tested the recoupling schemes based on the symmetry $SC4_2^0$, corresponding to the $[C4_2^0]_0[C4_2^0]_{120}[C4_2^0]_{240}$ sequence with a basic element $90_{45}90_{135}90_{45}90_{225}90_{315}90_{225}$, or $C6_3^0$ built from $90_{30}90_{120}90_{30}90_{240}90_{330}90_{240}$. These recoupling schemes, which have been recently proposed,(Perras et al., 2019) derive from the $SR4_1^2(90_{-45}90_{45}90_{-45})$ and $R12_3^5(90_{-45}90_{45}90_{-45})$ schemes.

As seen in Table 5, the sequences yielding the highest transfer efficiencies are by decreasing order RINEPT-CWc with $SR4_1^2$(tt)

or $R12_3^5$ (tt) > PRESTO- $R22_2^7$ ($180_0$) > RINEPT-CWc- $SR4_1^2$ ($270_090_{180}$) ≈ PRESTO- $R18_2^5$ ($180_0$) > RINEPT-CWc-$R12_3^5(270_090_{180})$. Figs.4 and 5 display the signal intensity of these sequences as function of the rf-field amplitude and offset, respectively.

The highest transfer efficiency is obtained with the RINEPT-CWc sequence incorporating an adiabatic recoupling. This recoupling also leads to the highest robustness to offset and rf inhomogeneity, and $SR4_1^2$(tt) and $R12_3^5$(tt) yield identical transfer

efficiency and robustness. Hence, the three-step multiple-quantum super-cycle of the $SR4_1^2$ symmetry does not improve the robustness in the case of a tanh/tan basic element. However, these recoupling schemes require maximum rf fields of $\nu_{1,max} \geq 8\nu_R = 160$ kHz, which may exceed the rf power specifications of most 3.2 mm MAS probes.

The PRESTO sequences using $R22_2^7(180_0)$ and $R18_2^5(180_0)$ recoupling also result in good transfer efficiencies, *i.e.*, 27 and 39 %, respectively, but lower than RINEPT-CWc-$SR4_1^2$(tt). However, they use rf-fields of $\nu_1/\nu_R = 5.5$ and 4.5, which are

compatible with the specifications of 3.2 mm MAS probes. The higher transfer efficiency of $R22_2^7(180_0)$ with respect to $R18_2^5(180_0)$ stems from its weaker second-order cross-terms between $^1$H-$^1$H interactions (Table 1).

The efficiency of the RINEPT-CWc-$SR4_1^2(270_090_{180})$ sequence, with rf-field $\nu_1 = 4\nu_R$, is comparable to that of PRESTO-$R18_2^5(180_0)$, but with a higher robustness to offset and rf inhomogeneity. We can notice that amplitude modulated recoupling schemes, for which the phase shifts are equal to 180°, such as $SR4_1^2(270_090_{180})$ and $SR4_1^2(180_0)$, exhibit a high robustness to

offset (Fig.5).(Carravetta et al., 2000) The use of $270_090_{180}$ composite pulses in $SR4_1^2$ symmetries instead of single π pulses improves their transfer efficiency as well as their robustness to offset and rf field inhomogeneity.

In summary, for $\nu_R = 20$ kHz in γ-alumina, the RINEPT-CWc-$SR4_1^2(270_090_{180})$ sequence achieves efficient and robust transfers of magnetization from protons to $^{27}$Al nuclei using a moderate rf field of $\nu_1 = 4\nu_R$. For $^1$H spectra with a width smaller than 20





kHz and MAS probes with a good rf-homogeneity, PRESTO-$R22_2^7(180_0)$ sequence can result in slightly higher transfer

efficiencies.

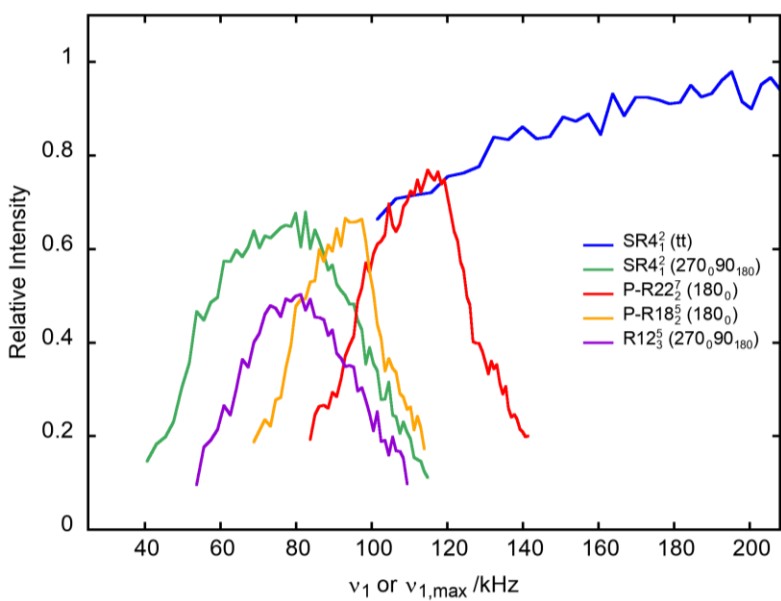

**Figure 4: Variation at $\nu_R$ = 20 kHz of the $^{27}AlO_6$ signal of $\gamma$-alumina as function of $\nu_1$ or $\nu_{1,max}$ of the recoupling for PRESTO-$R22_2^7(180_0)$ and $-R18_2^5(180_0)$ as well as RINEPT-$SR4_1^2$ (tt), $-SR4_1^2$ ($270_090_{180}$) and $-R12_3^5$ ($270_090_{180}$). For each curve $\tau$ was fixed to its optimum value given in Table 5.**

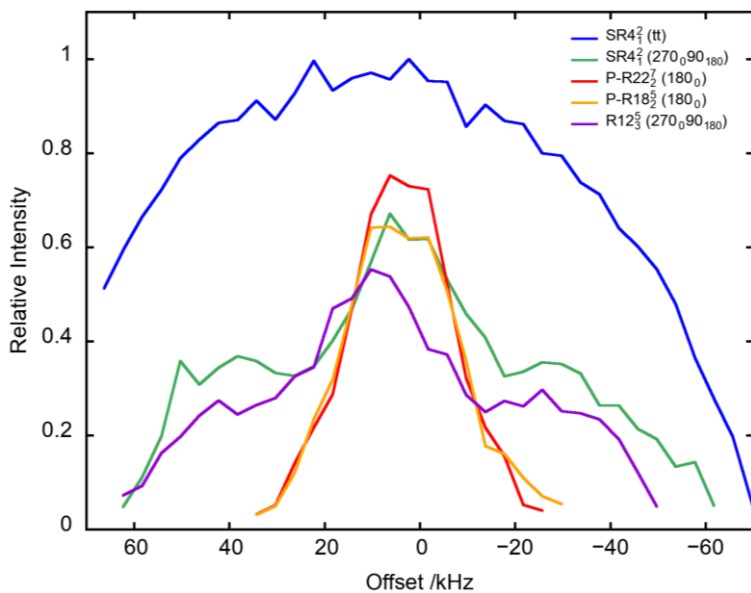


**Figure 5: Variation at $\nu_R$ = 20 kHz of the $^{27}AlO_6$ signal of $\gamma$-alumina as function of offset of the recoupling for PRESTO-$R22_2^7(180_0)$ and $-R18_2^5(180_0)$ as well as RINEPT-$SR4_1^2$ (tt), $-SR4_1^2$ ($270_090_{180}$) and $-R12_3^5$ ($270_090_{180}$). For each curve $\tau$ and $\nu_1$ or $\nu_{1,max}$ were fixed to their optimum values given in Table 5.**



**Table 5. Comparison of the performances of $^{1}H \rightarrow {}^{27}Al$ RINEPT-CWc and PRESTO transfers using various recoupling for AlO$_6$ signal of γ-alumina at $\nu_R$ = 20 kHz.**

| PRESTO /RINEPT | Recoupling | $\tau$ /μs | $\nu_1/\nu_{1,max}$ /kHz | AlO$_6$ [a] | $\Delta\nu_0$ [b] /kHz | $\Delta\nu_0/\nu_1$ | $\Delta\nu_1$ [c] /kHz | $\Delta\nu_1/\nu_1$ |
|---|---|---|---|---|---|---|---|---|
| RINEPT | $SR4_1^2(tt)$ | 400 | 160 | 1 | 110 | 0.68 | > 100[d] | > 0.62 |
|  | $R12_3^5(tt)$ | 400 | 160 | 1 | 110 | 0.68 | > 100[d] | > 0.62 |
| PRESTO | $R22_2^7(180_0)$ | 400 | 110 | 0.73 | 30 | 0.27 | 39 | 0.35 |
| RINEPT | $SR4_1^2(270_0 90_{180})$ | 400 | 80 | 0.63 | 50 | 0.63 | 44 | 0.55 |
| PRESTO | $R18_2^5(180_0)$ | 400 | 90 | 0.61 | 28 | 0.31 | 27 | 0.30 |
|  | $R12_3^5(270_0 90_{180})$ | 400 | 80 | 0.50 | 40 | 0.50 | 35 | 0.44 |
|  | $SR4_1^2(90_{-45} 90_{45} 90_{-45})$ | 400 | 63 | 0.42 | 14 | 0.22 | 14 | 0.22 |
|  | $SR4_1^2(180_0)$ | 400 | 45 | 0.40 | 17 | 0.38 | 24 | 0.53 |
| RINEPT | $R12_3^5(180_0)$ | 400 | 45 | 0.35 | 10 | 0.22 | 15 | 0.33 |
|  | $R12_3^5(90_{-45} 90_{45} 90_{-45})$ | 400 | 66 | 0.35 | 11 | 0.17 | 18 | 0.27 |
|  | $SC2_1^0$ | 400 | 63 | 0.31 | 14 | 0.22 | 45 | 0.71 |
|  | $C6_3^0$ | 400 | 66 | 0.28 | 10 | 0.15 | 40 | 0.60 |

[a] AlO$_6$ signal normalized to that with $^{1}H \rightarrow {}^{27}Al$ RINEPT-CWc-$SR4_1^2(tt)$. [b] FWHM of the robustness to offset. [c] FWHM of the robustness to rf-field. [d] Only a lower bound of rf-field could be determined due to probe rf specifications (Fig.4).

**IV-3-2. Isopropylamine-templated AlPO$_4$-14**

Fig.6 shows the 1D $^{1}H \rightarrow {}^{27}Al$ RINEPT and PRESTO spectra of AlPO$_4$-14. They exhibits three $^{27}Al$ resonances at 43, 21 and −2 ppm assigned to AlO$_4$, AlO$_5$ and AlO$_6$ sites, respectively.(Ashbrook et al., 2008) The AlO$_5$ and AlO$_6$ sites are directly bonded to OH groups. The $^{1}H$ MAS spectrum is shown in Fig.S1. According to the literature, the $^{27}AlO_4$ signal subsumes the resonances of two AlO$_4$ sites with quadrupolar coupling constants $C_Q$ = 1.7 and 4.1 MHz, whereas the $C_Q$ constants of $^{27}AlO_5$ and $^{27}AlO_6$ sites are equal to 5.6 and 2.6 MHz, respectively.(Fernandez et al., 1996; Antonijevic et al., 2006) The $^{1}H$-$^{1}H$ dipolar couplings within the isopropylamine template molecule are larger than in γ-alumina. We used the most intense peak, AlO$_4$, to compare the $^{1}H \rightarrow {}^{27}Al$ transfer efficiencies of RINEPT-CWc and PRESTO sequences with different recoupling schemes, and the results are given in Table 6. The six sequences yielding the highest transfer efficiencies are the same for AlPO$_4$-14 and γ-alumina and their relative efficiencies are comparable for the AlO$_4$ peak of AlPO$_4$-14 and the AlO$_6$ signal of γ-alumina.



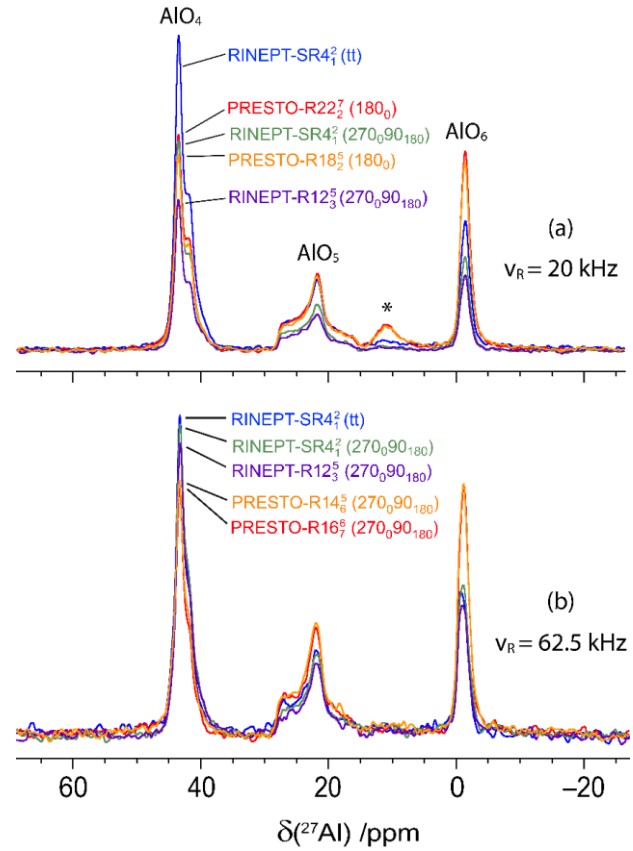

**Figure 6: 1D $^{27}$Al spectra of AlPO$_4$-14 at B$_0$ = 18.8 T with $\nu_R$ = 20 (a) and 62.5 (b) kHz acquired using $^1$H → $^{27}$Al RINEPT-CWc and PRESTO-III transfers using the following recoupling schemes: SR4$_1^2$(tt), SR4$_1^2$(270$_0$90$_{180}$) and R12$_3^5$(270$_0$90$_{180}$) for RINEPT, and (a) R22$_2^7$(180$_0$) and R18$_2^5$ (180$_0$), or (b) R16$_7^6$ (270$_0$90$_{180}$) and R14$_6^5$ (270$_0$90$_{180}$) for PRESTO. $\tau$ delay and $\nu_1/\nu_{1,max}$ rf field were fixed to their optimal values given in Tables 6 and 8. The resonance at *ca*. 11 ppm in (a) is due to an impurity.**

Nevertheless, the rf requirement of the SR4$_1^2$(tt) and R12$_3^5$(tt) schemes is higher for AlPO$_4$-14 than for γ-alumina because of larger $^1$H-$^1$H dipolar couplings, in agreement with the numerical simulations of Figs.2a-c. This rf requirement prevents the use of these adiabatic recoupling schemes at $\nu_R$ = 20 kHz with most 3.2 mm MAS probes. The rf requirement of the other sequences, and their robustness to offset and rf-fields homogeneity are similar for both samples (Table 6 and Figs.S2 and S3). With respect to the RINEPT sequence, PRESTO yields higher transfer efficiency for AlO$_5$ and AlO$_6$ resonances than for AlO$_4$ signals in the case of AlPO$_4$-14 since (i) these Al sites are directly bonded to OH groups and (ii) R22$_2^7$(180$_0$) and R18$_2^5$(180$_0$) recoupling schemes are subject to dipolar truncation (section II-1-1), which prevents to transfer the $^1$H magnetization of these OH groups to $^{27}$AlO$_4$ nuclei.

Hence, for both AlPO$_4$-14 and γ-alumina, RINEPT-CWc-SR4$_1^2$(270$_0$90$_{180}$) and PRESTO-R22$_2^7$(180$_0$) sequences are the methods of choice to transfer $^1$H magnetization to $^{27}$Al nuclei at $\nu_R$ = 20 kHz.



**Table 6. Comparison of the performances of $^1H \rightarrow {}^{27}Al$ RINEPT-CWc and PRESTO transfers with AlPO$_4$-14 at $\nu_R$ = 20 kHz.**

| PRESTO /RINEPT | Recoupling | $\tau$ /$\mu$s | $\nu_1/\nu_{1,max}$ /kHz | Intensity [a] | | | $\Delta\nu_0$ /kHz | $\Delta\nu_0/\nu_1$ | $\Delta\nu_1$ /kHz | $\Delta\nu_1/\nu_1$ |
|---|---|---|---|---|---|---|---|---|---|---|
| | | | | AlO$_6$ | AlO$_5$ | AlO$_4$ | | | | |
| RINEPT | $SR4_1^2(tt)$ | 800 | 208 | 1 | 1 | 1 | 120 | 0.58 | - [b] | - [b] |
| | $R12_3^5(tt)$ | 800 | 208 | 0.99 | 0.99 | 0.98 | 120 | 0.58 | - [b] | - [b] |
| PRESTO | $R22_2^7(180_0)$ | 600 | 114 | 1.54 | 1.07 | 0.67 | 26 | 0.23 | 38 | 0.33 |
| RINEPT | $SR4_1^2(270_090_{180})$ | 800 | 77 | 0.72 | 0.65 | 0.67 | 45 | 0.58 | 48 | 0.62 |
| PRESTO | $R18_2^5(180_0)$ | 600 | 94 | 1.45 | 1.03 | 0.62 | 25 | 0.27 | 26 | 0.28 |
| RINEPT | $R12_3^5(270_090_{180})$ | 800 | 77 | 0.58 | 0.50 | 0.48 | 46 | 0.60 | 36 | 0.47 |
| | $SR4_1^2(180_0)$ | 600 | 43 | 0.64 | 0.45 | 0.36 | 14 | 0.33 | 23 | 0.53 |
| | $SR4_1^2(90_{-45}90_{45}90_{-45})$ | 800 | 61 | 0.56 | 0.43 | 0.25 | 16 | 0.26 | 20 | 0.32 |
| | $SC2_1^0$ | 800 | 68 | 0.54 | 0.41 | 0.24 | 18 | 0.26 | 52 | 0.73 |
| | $R12_3^5(90_{-45}90_{45}90_{-45})$ | 600 | 61 | 0.43 | 0.30 | 0.21 | 8 | 0.13 | 18 | 0.29 |
| | $R12_3^5(180_0)$ | 600 | 45 | 0.34 | 0.28 | 0.21 | 8 | 0.18 | 18 | 0.40 |
| | $C6_3^0$ | 600 | 68 | 0.52 | 0.36 | 0.21 | 10 | 0.15 | 42 | 0.61 |

[a] Intensities of AlO$_6$, AlO$_5$ and AlO$_4$ resonances normalized to their intensities with $^1H \rightarrow {}^{27}Al$ RINEPT-CWc-$SR4_1^2(tt)$.
[b] FWHM of the robustness to rf-field was not measured for RINEPT-$SR4_1^2(tt)$ and -$R12_3^5(tt)$ (Fig.S1).


## IV-4. PRESTO and RINEPT performances for $\nu_R$ = 62.5 kHz

Similar comparisons of the performances of the various RINEPT-CWc and PRESTO sequences were performed for $\gamma$-alumina and AlPO$_4$-14 at $\nu_R$ = 62.5 kHz.

### IV-4-1. $\gamma$-alumina

The corresponding data for $\gamma$-alumina are given Table 7. The sequences yielding the highest transfer efficiencies are by decreasing order RINEPT-CWc with $SR4_1^2(tt)$ or $R12_3^5(tt)$ > RINEPT-CWc-$SR4_1^2(270_090_{180})$ ≈ PRESTO-$R16_7^6(270_090_{180})$ > PRESTO-$R14_6^5(270_090_{180})$ > RINEPT-CWc-$R12_3^5(270_090_{180})$.

Nevertheless, the nominal rf requirements of the RINEPT sequences using adiabatic pulses or $270_090_{180}$ composite $\pi$-pulses correspond to $\nu_{1max}$ = 5$\nu_R$ (313 kHz: Fig.2d) or 4$\nu_R$ (250 kHz), which exceeds the specifications of our 1.3 MAS probe, and the

sequences were tested only up to $\nu_{1max}$ = 208 kHz (Fig.7). This suboptimal rf field could potentially limit the transfer efficiencies of these sequences.

The PRESTO-$R16_7^6(270_090_{180})$ and -$R14_6^5(270_090_{180})$ sequences yield transfer efficiencies comparable to those of RINEPT-CWc-$SR4_1^2(270_090_{180})$, but with a significantly lower rf field, 137 kHz ≈ 2.3$\nu_R$. Furthermore, the robustness to offset of these



PRESTO sequences is comparable to that of RINEPT-CWc-SR4$_1^2$(270$_0$90$_{180}$) (Fig.8). PRESTO-R22$_4^3$(180$_0$) and -R16$_3^2$(180$_0$)

sequences with the small phase shift of $2\phi \leq 52°$ are less efficient because they are sensitive to rf inhomogeneity.

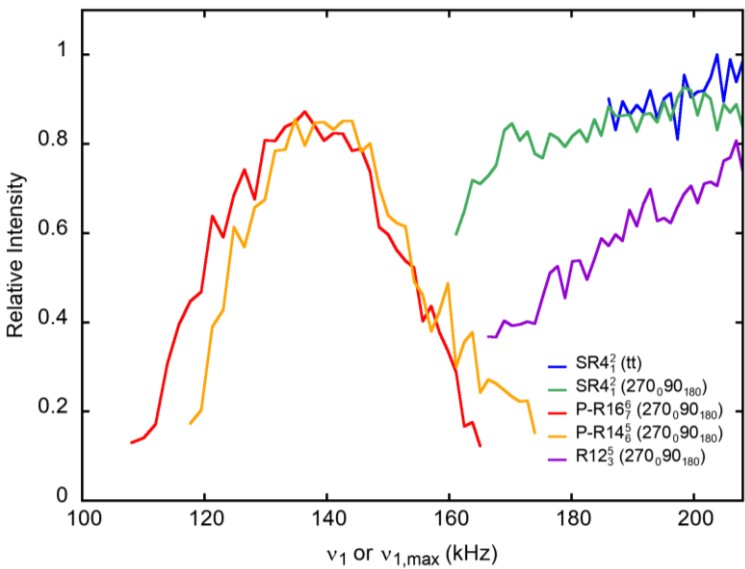

**Figure 7: Variation at $\nu_R$ = 62.5 kHz of the $^{27}$AlO$_6$ signal of γ-alumina as function of $\nu_1$ or $\nu_{1,max}$ of the recoupling for PRESTO-R16$_7^6$(270$_0$90$_{180}$) and -R14$_6^5$(270$_0$90$_{180}$) as well as RINEPT-SR4$_1^2$ (tt), -SR4$_1^2$ (270$_0$90$_{180}$) and -R12$_3^5$ (270$_0$90$_{180}$). For each curve $\tau$ was**
**fixed to its optimum value given in Table 7.**

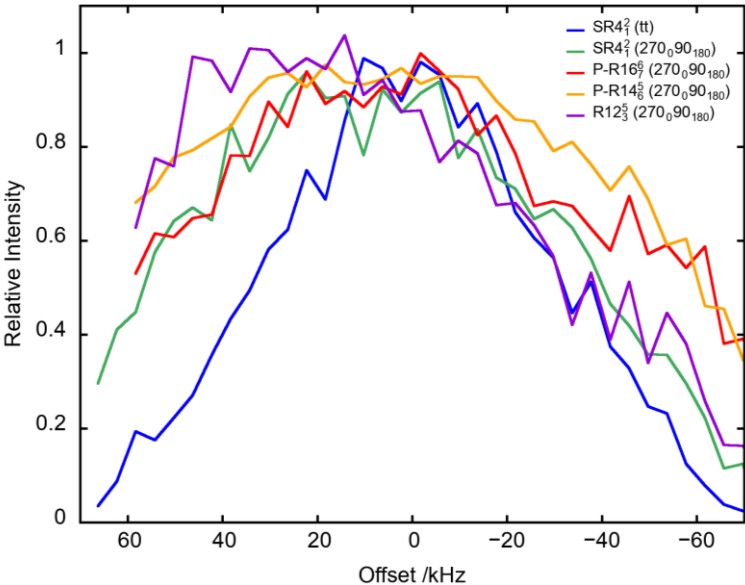

**Figure 8: Variation at $\nu_R$ = 62.5 kHz of the $^{27}$AlO$_6$ signal of γ-alumina as function of offset of the recoupling for PRESTO-R16$_7^6$(270$_0$90$_{180}$) and -R14$_6^5$(270$_0$90$_{180}$) as well as RINEPT-SR4$_1^2$ (tt), -SR4$_1^2$ (270$_0$90$_{180}$) and -R12$_3^5$ (270$_0$90$_{180}$). For each curve $\tau$ and $\nu_1$ or $\nu_{1,max}$ were fixed to their optimum values given in Table 7.**



**Table 7. Comparison of the performances of $^1H\rightarrow ^{27}Al$ RINEPT-CWc and PRESTO transfer using various recoupling for the AlO$_6$ signal of γ-alumina at ν$_R$ = 62.5 kHz.**

| PRESTO/RINEPT | Recoupling | $\tau$ /μs | $\nu_1/\nu_{1,max}$ /kHz | AlO$_6$ [a] | $\Delta\nu_0$ /kHz | $\Delta\nu_0/\nu_1$ | $\Delta\nu_1$ /kHz | $\Delta\nu_1/\nu_1$ |
|---|---|---|---|---|---|---|---|---|
| RINEPT | $SR4_1^2(tt)$ | 256 | 208 | 1 | 74 | 0.36 | -[b] | -[b] |
| | $R12_3^5(tt)$ | 256 | 208 | 1 | 74 | 0.36 | -[b] | -[b] |
| | $SR4_1^2(270_090_{180})$ | 320 | 208 | 0.92 | 96 | 0.46 | -[b] | -[b] |
| PRESTO | $R16_7^6(270_090_{180})$ | 448 | 137 | 0.91 | 90 | 0.66 | 42 | 0.31 |
| | $R14_6^5(270_090_{180})$ | 384 | 146 | 0.86 | 100 | 0.68 | 38 | 0.26 |
| RINEPT | $R12_3^5(270_090_{180})$ | 320 | 208 | 0.82 | 86 | 0.41 | -[b] | -[b] |
| | $SR4_1^2(180_0)$ | 320 | 125 | 0.75 | 52 | 0.42 | 88 | 0.70 |
| | $R12_3^5(180_0)$ | 288 | 125 | 0.74 | 16 | 0.13 | 85 | 0.68 |
| PRESTO | $R22_4^3(180_0)$ | 256 | 157 | 0.67 | 68 | 0.43 | 20 | 0.13 |
| | $R16_3^2(180_0)$ | 384 | 155 | 0.51 | 48 | 0.31 | 40 | 0.26 |
| RINEPT | $SC2_1^0$ | 256 | 186 | 0.34 | 50 | 0.27 | 84 | 0.45 |
| | $C6_3^0$ | 256 | 186 | 0.34 | 43 | 0.23 | 76 | 0.41 |
| | $SR4_1^2(90_{-45}90_{45}90_{-45})$ | 256 | 186 | 0.32 | 47 | 0.25 | 70 | 0.38 |
| | $R12_3^5(90_{-45}90_{45}90_{-45})$ | 256 | 186 | 0.32 | 40 | 0.22 | 70 | 0.38 |

[a] Intensities of AlO$_6$, AlO$_5$ and AlO$_4$ resonances normalized to their intensities with $^1H \rightarrow ^{27}Al$ RINEPT-CWc-$SR4_1^2(tt)$.
[b] FWHM of the robustness to rf-field was not measured for RINEPT-$SR4_1^2(tt)$ and -$R12_3^5(tt)$ (Fig.7).

**IV-4-2. Isopropylamine-templated AlPO$_4$-14**

In the case of AlPO$_4$-14, the relative transfer efficiencies for $^{27}AlO_4$ nuclei follow a similar order as for γ-alumina, except that the transfer efficiencies of PRESTO-$R16_7^6(270_090_{180})$ and -$R14_6^5(270_090_{180})$ are significantly lower than that of RINEPT-CWc-$SR4_1^2(270_090_{180})$ (Table 8). This decreased efficiency of the PRESTO schemes for AlO$_4$ stems notably from the dipolar truncation, which prevents the transfer of magnetization from the protons of OH groups bonded to AlO$_5$ and AlO$_6$ sites to $^{27}AlO_4$ nuclei. In Table. S1 we give the $^1H$-$^{27}Al$ distances of AlPO$_4$-14, which confirms that OH groups are closer to AlO$_5$ and AlO$_6$ sites, and hence, why the transfer efficiency for PRESTO-$R16_7^6(270_090_{180})$ and -$R14_6^5(270_090_{180})$ is higher than for RINEPT-CWc-$SR4_1^2(270_090_{180})$. However, the latter sequence uses amplitude-modulated recoupling, and hence benefits from a higher robustness to rf-field inhomogeneity than the PRESTO schemes (Fig.S3). Conversely, the robustness to offset of these three sequences are comparable (Fig.S4), whereas the rf requirements of $R16_7^6(270_090_{180})$ and $R14_6^5(270_090_{180})$ are much lower than that of $SR4_1^2(270_090_{180})$.



In summary, at $\nu_R = 62.5$ kHz, for both γ-alumina and isopropylamine-templated AlPO$_4$-14, PRESTO-R16$_7^6$(270$_0$90$_{180}$) and RINEPT-CWc-SR4$_1^2$(270$_0$90$_{180}$) are the methods of choice to transfer polarization of protons to quadrupolar nuclei. However, the first sequence requires a much lower rf-field than the second.

**Table 8. Comparison of the performances of $^1$H → $^{27}$Al RINEPT-CWc and PRESTO transfers using various recoupling for AlPO$_4$-14 at $\nu_R = 62.5$ kHz.**

| PRESTO /RINEPT | Recoupling | $\tau$ /μs | $\nu_1/\nu_{1,max}$ /kHz | Intensity [a] | | | $\Delta\nu_0$ /kHz | $\Delta\nu_0/\nu_1$ | $\Delta\nu_1$ /kHz | $\Delta\nu_1/\nu_1$ |
|---|---|---|---|---|---|---|---|---|---|---|
| | | | | AlO$_6$ | AlO$_5$ | AlO$_4$ | | | | |
| RINEPT | SR4$_1^2$(tt) | 480 | 208 | 1 | 1 | 1 | 48 | 0.23 | - [b] | - [b] |
| | R12$_3^5$(tt) | 480 | 208 | 1.07 | 1 | 1.06 | 44 | 0.21 | - [b] | - [b] |
| | SR4$_1^2$(270$_0$90$_{180}$) | 480 | 208 | 1.05 | 0.95 | 0.97 | 85 | 0.41 | 90 | 0.43 |
| | R12$_3^5$(270$_0$90$_{180}$) | 480 | 208 | 0.91 | 0.84 | 0.91 | 80 | 0.38 | 68 | 0.33 |
| PRESTO | R16$_7^6$(270$_0$90$_{180}$) | 672 | 146 | 1.71 | 1.21 | 0.76 | 80 | 0.55 | 50 | 0.34 |
| | R14$_6^5$(270$_0$90$_{180}$) | 576 | 146 | 1.72 | 1.27 | 0.76 | 86 | 0.59 | 45 | 0.31 |
| RINEPT | SR4$_1^2$(180$_0$) | 480 | 129 | 0.84 | 0.79 | 0.75 | 48 | 0.37 | 64 | 0.49 |
| | R12$_3^5$(180$_0$) | 480 | 136 | 0.72 | 0.67 | 0.74 | 18 | 0.13 | 54 | 0.40 |
| PRESTO | R22$_4^3$(180$_0$) | 512 | 157 | 1.47 | 1.18 | 0.69 | 60 | 0.38 | 20 | 0.33 |
| | R16$_3^2$(180$_0$) | 480 | 147 | 1.17 | 0.83 | 0.52 | 64 | 0.44 | 20 | 0.31 |
| RINEPT | R12$_3^5$(90$_{-45}$90$_{45}$90$_{-45}$) | 256 | 190 | 0.48 | 0.27 | 0.14 | 32 | 0.17 | 75 | 0.39 |
| | C6$_3^0$($C$') | 256 | 193 | 0.47 | 0.28 | 0.14 | 28 | 0.15 | 78 | 0.40 |
| | SR4$_1^2$(90$_{-45}$90$_{45}$90$_{-45}$) | 256 | 196 | 0.48 | 0.14 | 0.14 | 36 | 0.18 | 77 | 0.39 |
| | SC2$_1^0$($C$) | 256 | 188 | 0.53 | 0.25 | 0.14 | 44 | 0,23 | 80 | 0.43 |

[a] Intensities of AlO$_6$, AlO$_5$ and AlO$_4$ resonances normalized to their intensities with $^1$H→$^{27}$Al RINEPT-CWc-SR4$_1^2$(tt).
[b] FWHM of the robustness to rf-field was not measured for RINEPT-SR4$_1^2$(tt) and -R12$_3^5$(tt) (Fig.S3).

## V. Conclusions

In this work, we have introduced novel symmetry-based hetero-nuclear dipolar recoupling schemes, which can be incorporated into the RINEPT and PRESTO sequences to transfer the magnetization from protons to half-integer quadrupolar nuclei at $\nu_R = 20$ or 62.5 kHz. These novel recouplings have been compared to existing schemes. We have shown that the RINEPT-CWc-SR4$_1^2$(tt) sequence, which produces efficient and robust transfers at $\nu_R \approx 10$-15 kHz,(Nagashima et al., 2020) requires rf-fields incompatible with the specifications of most MAS probes for $\nu_R \geq 20$ kHz. Conversely, the introduced RINEPT-CWc-
SR4$_1^2$(270$_0$90$_{180}$) and PRESTO-R22$_2^7$(180$_0$) techniques with rf-fields of $4\nu_R$ and $5.5\nu_R$, respectively, are the methods of choice

**MAGNETIC RESONANCE**
Discussions
at $\nu_R$ = 20 kHz to transfer the magnetization from protons to quadrupolar nuclei. At $\nu_R$ = 62.5 kHz, the RINEPT-CWc-SR4$_1^2$(270$_0$90$_{180}$) and PRESTO-R16$_7^6$(270$_0$90$_{180}$) sequences with rf-requirements of 4$\nu_R$ and 2.3$\nu_R$, respectively, result in the most robust and efficient transfers. At both MAS frequencies, the RINEPT and PRESTO techniques complement each other since the latter is dipolar truncated, whereas the former is not. As result, the RINEPT sequences must be chosen to observe
simultaneously protonated and unprotonated sites, whereas the PRESTO schemes can be employed for the selective observation of quadrupolar nuclei in proximity to protons. These techniques are expected to be useful for transferring the DNP-enhanced magnetization of protons to quadrupolar nuclei in indirect MAS DNP experiments at $\nu_R \geq$ 20 kHz, notably used at high magnetic fields.(Nagashima et al., 2020, n.d.; Rankin et al., 2019; Berruyer et al., 2020)

**Author contributions**: JSG, AGMR and JT carried out the NMR experiments on γ-alumina and AlPO$_4$-14. YT performed the spin dynamics simulations and carried out the NMR experiments on l-histidine·HCl. OL derived average Hamiltonian theory for the investigated recoupling sequences. OL and JPA wrote the manuscript. All the authors contributed to the editing of the manuscript.

**Acknowledgments**

This article is dedicated to Dr Francis Taulelle, our friend, who passed away very recently. The Chevreul Institute (FR 2638), Ministère de l'Enseignement Supérieur, de la Recherche et de l'Innovation, Hauts-de-France Region, and FEDER are acknowledged for supporting and funding partially this work. Financial support from the IR-RMN-THC FR-3050 CNRS for conducting the research is gratefully acknowledged. This project has received funding from the European Union's Horizon 2020 research and innovation program under grant agreement No. 731019 (EUSMI). OL acknowledges financial support from
Institut Universitaire de France (IUF) and contract ANR-18-CE08-0015-01 (ThinGlass). FP acknowledges financial support from I-site contract OPE-2019-0043 (5400-MOFFIN).

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
