# Peer review of "Improved NMR transfer of magnetization from protons to halfinteger spin quadrupolar nuclei at moderate and high MAS frequencies"

_Magnetic Resonance, 2021_

## Author Response (AR1)

I found this to be a very nice paper by Gomez et al. comparing the efficiency of SQ and ZQ symmetry-based polarization transfer sequences at moderate and high MAS rates. The search for optimal sequences was thorough, which adds confidence in the results. I have a few minor corrections and comments that I will outline below.

1) Line 99 – Although the sentence is correct, it sounds as though the author is trying to say that the rf power increasing with homo-nuclear dipolar interactions makes it unsuitable for high MAS rates. I think it should be split into two sentences.
As suggested by the referee, this sentence has been split: "We demonstrate using numerical simulations of spin dynamics and experiments on γ-alumina and isopropylamine templated microporous aluminophosphate $AlPO_4$-14 (hereafter $AlPO_4$-14) that the rf requirement of this technique increases with the $^1H$-$^1H$ dipolar interactions. In practice, this rf requirement is not compatible with the specifications of most MAS probes at $\nu_R \geq 20$ kHz, even for moderate $^1H$-$^1H$ dipolar interactions."

2) Line 112 – Why define R'? The phase $\phi$ is defined and R with a phase of -$\phi$ is sufficient.
We kept the notation $\mathcal{R}'$, which has been widely used in publications about $RN_n^\nu$ sequences (see Carravetta et al 2000, Brinkmann et al 2001 and 2004, for instance). We added the following sentence in the first paragraph of subsection II-1-1: "$\mathcal{R}$ and $\mathcal{R}'$ are identical when they are amplitude-modulated, i.e. all phase shifts are multiple of $\pi$." Indeed, for the basic element $\mathcal{R} = 90_0 240_{90} 90_0$, $\mathcal{R}' = 90_0 240_{-90} 90_0$ differs from $\mathcal{R}$.

3) Line 137 – Please include a reference for dipolar truncation.

The reference Bayro et al 2009 about dipolar truncation has been cited.

4) Line 164. The reason given for screening the phase of the sequence to those near 90° is that it is what is used, but the authors could be more specific that these sequences are better compensated for rf inhomogeneity due to the 180° phase difference between the two pulses.
The sentence has been modified into "We only considered the $RN_n^\nu$ symmetries with $45 \leq \phi \leq 135°$ since sequences with $\phi$ close to 90° are better compensated for rf field errors and inhomogeneities (Brinkmann and Kentgens, 2006b).".

5) Line 264 – This statement may be misleading. $R12_9^5$ has a non-zero scaling factor when the rf field is greater than $2\nu_R$.
We agree with the reviewer. The sentence has been modified into: "Contrary to the $RN_n^\nu$ with $|m| = 2$ SQ hetero-nuclear dipolar recouplings, the rf-field of the $RN_n^\nu$ with $|m| = 2$ two-spin order schemes is always higher than $2\nu_R$ since these symmetries with $2n > N$, such as $R12_9^5$, have smaller $\kappa$ scaling factors for the basic elements employed here."

6) Figs.2e and f are distorted. The figures have been modified.

7) My last comment has to do with the generality of the results being presented and perhaps the authors might want to look into this point more deeply. As the authors have demonstrated in an earlier JACS paper, the INEPT sequences can outclass PRESTO when the dipolar couplings are weak, for instance with low-γ nuclei, since the ZQ sequences can sustain longer recoupling times. We have also noticed that PRESTO-II far surpasses INEPT when the dipolar couplings are very strong (since it enables for very short non-synchronized recoupling periods, presumably), and so there might be some cross-over point in dipolar coupling strength when INEPT surpasses PRESTO. $^{27}Al$ is likely near this cross-over point. So likely, the best RINEPT sequences will remain the best,

and the same for the PRESTO sequences, but the comparison of the two types of sequences may be very dependent on the dipolar coupling strength and the time required to recouple the interaction. We have recently shown that for large $^1$H-$^{17}$O dipolar couplings, a variant of the RINEPT sequence with two $^{17}$O pulses is more efficient than PRESTO-II (see Nagashima et. al, *Magnetic Resonance in Chemistry*, In press, https://doi.org/10.1002/mrc.5121). The following sentence has been inserted in the introduction: "Furthermore, for quadrupolar nuclei subject to large dipolar interactions, such as $^{17}$O nuclei of OH group, we have shown that a RINEPT-CWc-SR4$_1^2$(tt) version with only two pulses on the quadrupolar channel is more efficient that its PRESTO counterpart (Nagashima et al., 2020).".
* * *
**Z. Gan.** There has been an increasing interest in CPMAS from $^1$H to half-integer quadrupolar nuclei in order to materialize the high $^1$H DNP enhancement through efficient spin diffusion for detecting quadrupolar nuclei. This work is practically useful to this effort in addition to the proximity information by optimizing the pulse sequences. The authors have chosen γ-alumina as the sample system, which has a fairly large shift dispersion at 800 MHz field. The proton homo-nuclear coupling is relatively weak as compared to rigid organic molecules. They have also chosen 20 and 62.5 kHz spinning speeds and typical rf-fields that 3.2 and 1.3 mm probes can achieve. From previous publications, the authors also have added adiabatic pulses to improve the offset performance and some modifications to the blank rotor period to reduce the $^1$H $T_2$' loss. There is no doubt that they have improved the performances, but I am not sure if this optimization is generally applicable to other samples and spin systems. My particular concern is the strong $^1$H homo-nuclear coupling. It is correct that the first-order $^1$H homo-nuclear term is not 'recoupled' for the selected hetero-nuclear recoupling sequences. What about the high-order homo-nuclear terms that cause the short $T_2$', especially at not-so-high spinning speeds? My observation is that $^1$H $T_2$' under the recoupling is often very short, and thus can have a strong effect onto the performances.

8) I wonder if the authors could provide some $^1$H $T_2$' information under the recoupling sequences, say as compared to a regular spin-echo $T_2$' decay.
As requested by Zhehong, we measured the $^1$H $T_2$' constants for AlPO$_4$-14 under the most efficient recoupling sequences. These constants are reported in Table 5 and discussed in section IV-5.

9) In addition, is it possible to program the experiment to avoid the blank rotor period to rotor during the CT π-pulse by simultaneous pulses.
The ZQ recoupling employed in RINEPT are non-γ-encoded. As the π-pulse on the $^1$H channel has a finite length, the two recoupling periods bracketing this π-pulse must be rotor-synchronized and the window delays cannot be avoided. This is notably shown in Fig.S3 of the reference Nagashima et a. 2021, we recently published.

10) I would be interested in the overall transfer efficiency. It seems very low based on the S/N. Of course, only a small portion of γ-alumina has $^1$H in its vicinity. It would help to know where the main loss comes from, for future improvement.
It is hard to measure experimentally the overall transfer efficiency for hetero-nuclear magnetization transfer. For γ-alumina, the low S/N stems from the small amount of protons in this sample and the use of 1.3 mm rotor. For AlPO$_4$-14, higher S/N was obtained, even if we still used a 1.3 mm rotor.

11) The first citation on PRESTO, p2 line-66, should include the paper from the original paper from Levitt's group.
As requested by the referee, the reference Zhao et al 2004 is now cited after the first citation of PRESTO technique.

I would like to recommend this work on the through description of symmetry based recoupling and search for recoupling sequences that are optimal for CPMAS with practically feasible spinning speed and rf field for 3.2 and 1.3 mm probes.